# Universal Learning of Nonlinear Dynamics

**Evan Dogariu** [1]  **Anand Brahmbhatt** [2]  **Elad Hazan** [2]

## Abstract

We study the fundamental problem of one-step prediction of a marginally stable unknown nonlinear dynamical system. We describe an algorithm for this problem, based on the technique of spectral filtering, which learns a mapping from past observations to the next based on a spectral representation of the system. Using techniques from online convex optimization, we prove vanishing prediction error for any nonexpansive nonlinear dynamical system with finitely many marginally stable modes, with rates governed by a novel quantitative control-theoretic notion of learnability. The main technical component of our method is a new spectral filtering algorithm for linear dynamical systems, which incorporates past observations and applies to general noisy and marginally stable systems. This generalizes the original spectral filtering algorithm to both asymmetric dynamics as well as incorporating noise correction, and is of independent interest.

## 1. Introduction

The problem of predicting a dynamical system from observations is a cornerstone of scientific and engineering inquiry. Across disciplines ranging from econometrics and meteorology to robotics, neuroscience, and recently language modeling, a central task is to predict the future evolution of a system given only a sequence of its past measurements. Formally, given a sequence of observations $(y_1, ..., y_t)$ generated by an unknown, underlying dynamical system of the form

$$x_{t+1} = f(x_t), \quad y_t = h(x_t), \quad (1.1)$$

the objective is to produce an accurate forecast $\hat{y}_{t+1}$. The challenge lies in the fact that the state $x_t$ is hidden and may

[1]New York University [2]Department of Computer Science, Princeton University. Correspondence to: Evan Dogariu <ed2719@nyu.edu>.

*Proceedings of the 43rd International Conference on Machine Learning*, Seoul, South Korea. PMLR 306, 2026. Copyright 2026 by the author(s).

be high-dimensional, and the dynamics function $f : \mathcal{X} \to \mathcal{X}$ and observation function $h : \mathcal{X} \to \mathcal{Y}$ are unknown.

Historically, two dominant paradigms have emerged to address this challenge. The first, rooted in control theory and system identification, aims to construct an explicit model of the underlying dynamics. This model-based approach seeks to learn the function $f$ or, more commonly, a linear representation of it. A powerful theoretical tool in this domain is the Koopman operator, which recasts the nonlinear dynamics of states into an infinite-dimensional, yet linear, evolution of observable functions. Data-driven methods such as Dynamic Mode Decomposition (DMD) attempt to find finite-dimensional approximations of this linear operator, yielding a model that can be used for prediction and control (Brunton et al., 2021). While elegant, these methods often rely on strong assumptions to guarantee the convergence of their approximations and can be computationally demanding.

The second paradigm, driven by recent advances in machine learning, employs powerful, general-purpose black-box architectures to learn the mapping from past to future observations directly. Architectures such as Transformers (Vaswani et al., 2017) and more recent state-space models like Mamba (Gu & Dao, 2023) and convolutional models like Hyena (Poli et al., 2023) have demonstrated remarkable success in sequence modeling tasks. These methods are highly expressive and often achieve state-of-the-art performance, but their operation can be opaque, and they typically lack the formal performance guarantees characteristic of control-theoretic approaches, especially in the presence of adversarial disturbances or for systems operating at the edge of stability.

**Provable improper learning for dynamical systems.**
This paper advocates for a third paradigm: improper learning for dynamical systems. The philosophy of this approach is that for the goal of prediction, it is not necessary to recover the true, and often intractably complex, underlying dynamics. Instead, one can design an efficient and provably correct algorithm by framing the learning problem as a competition. The algorithm does not attempt to learn a model from the same class as the true system (hence, "improper"); rather, it competes in hindsight against a class of idealized, well-behaved, and computationally tractable predictors.

This reframing of the problem is a key conceptual leap. It shifts the focus from the difficult, nonconvex problem of system identification to the more manageable, often convex, problem of regret minimization against a carefully chosen comparator class. The central idea of this work is to leverage the rich mathematical structure of control theory not for designing a controller, but for defining this very comparator class. Instead of attempting to learn the true nonlinear system, we design an algorithm that is guaranteed to perform as well as the best possible high-dimensional linear observer system for the given observation sequence. This approach allows us to sidestep the complexities of nonlinear system identification while retaining strong theoretical guarantees. Furthermore, our analysis yields a natural quantitative description of the complexity of learning from observations with improper algorithms, measured via spectral properties of approximate global linearizations.

**Our contributions.** This work makes several contributions to the theory and practice of learning dynamical systems, bridging concepts from classical control theory and modern machine learning.

1. We introduce **Observation Spectral Filtering** (OSF), an efficient algorithm based on online convex optimization that is guaranteed to learn any observable nonlinear dynamical system with finitely many marginally stable modes. The algorithm is "improper" in that it does not explicitly estimate the system's hidden states or dynamics; instead, it directly constructs a mapping from past observations to future predictions. This may be performed over raw observations or lifted features, yielding a flexible paradigm that works well empirically.

2. An **LDS guarantee:** The core technical engine of OSF is a learning guarantee for linear dynamical systems (LDS). This method applies to asymmetric dynamics under adversarial noise with performance guarantees that are independent of the system's hidden dimension. This generalizes prior spectral filtering methods, which were largely restricted to symmetric systems, and aligns our work with recent efforts to extend these powerful techniques to broader system classes.

3. **A control-theoretic analysis:** We provide a novel analytical framework that connects the predictability of a nonlinear system to the spectral properties of its best possible high-dimensional linear observer. This connection is formalized through a "Luenberger program," an optimization problem whose solution quantifies the inherent difficulty of learning the system from its outputs. The optimal value of this program, a condition number we denote $Q_\star$, appears directly in our data-dependent learning bounds. This framework estab-

lishes a quantitative link between the control-theoretic concept of observability and the machine learning concept of learnability, and provides an elegant analytic toolkit with which to derive learning guarantees in nonlinear settings using linear methods. We discuss further implications in Section 6.

4. **A simple global linearization:** We introduce a simple construction to approximate any bounded, nonexpansive nonlinear system with a high-dimensional LDS. This technique, based on state-space discretization, provides a rigorous foundation for applying our linear systems algorithm to the nonlinear case. It trades an increase in dimensionality — to which our learning algorithm is immune — for guaranteed linearity and approximation accuracy, and it uses the improper nature of our analysis to circumvent the strong spectral assumptions and convergence difficulties associated with many contemporary data-driven Koopman operator methods.

Ultimately, this work presents a synthesis of ideas from two distinct fields. It repurposes tools from control theory (such as the Luenberger observer, pole placement, and global linearization) not for their traditional purpose of observer/controller design, but as analytical instruments to define a robust comparator class for a learning algorithm. By then applying the machinery of online convex optimization, we develop an algorithm whose performance guarantees are directly informed by the structural properties of the underlying prediction problem. This combination of a constructive global linearization and spectral filtering yields a universal prediction algorithm for any nonexpansive nonlinear dynamical system with finitely many marginally stable modes, with regret scaling determined only by the system's robust observability constant $Q_\star$.

## 2. Related Work

In this section, we survey the past and current methods for sequence prediction in dynamical systems, highlighting their relationships, strengths, and limitations. While it is impossible to provide an exhaustive account, we highlight the works most relevant to our study.

### 2.1. Methods for Learning in Nonlinear Systems

Learning in nonlinear dynamical systems of the form (1.1), also called time series prediction in statistics, spans several fields of study. We can roughly divide the existing methods as follows:

**Statistical and online learning methods.** The classical text of Box and Jenkins (Box & Jenkins, 1976) introduces ARMA (autoregressive moving average) statistical estima-

tion techniques and their extensions, notably ARIMA. These methods are ubiquitous in applications and were extended to the adversarial online learning domain in (Anava et al., 2013; Kuznetsov & Mohri, 2018) via online convex optimization (Hazan, 2016). Later extensions include nonlinear autoregressive methods augmented with kernels, e.g. (Cuturi & Doucet, 2011). Also prominent in the statistical literature are nonparameteric time series methods such as Gaussian processes (Rasmussen, 2003; Murphy, 2012).

**Deep learning methods.** The most successful methods for learning dynamical systems are based on deep learning. Transformers (Vaswani et al., 2017) are the most widely used architectures that form the basis for large language models. More recent deep architectures are based on convolutional models such as Hyena (Poli et al., 2023) and spectral architectures such as FlashSTU (Liu et al., 2024). Recurrent neural networks and state space models such as Mamba (Gu & Dao, 2023) exhibit faster inference and require lower memory, albeit being harder to train and potentially less expressive.

**Koopman operator methods.** The most relevant technique for learning nonlinear dynamical systems is via the Koopman operator. This methodology relies on the mathematical fact that any (partially observed) dynamical system can be lifted to a linear representation in an infinite-dimensional space. The basic idea is to represent the evolution of the system by describing the evolution of all possible system observables via the Koopman operator, which becomes linear and allows for functional-analytic and spectral methods; see (Mezic, 2020) for a survey of the related theory. Detailed mathematical study of this operator can require complicated analysis and strong assumptions; a simple version of a related global linearization can be achieved via discretization, which we present in Section B.3.

Data-driven methods for Koopman operator learning are based on learning an approximate representation of the dynamics in high-dimensional linearly-evolving coordinates. Knowing the top $N$ eigenvalues and eigenfunctions of the Koopman operator allows one to approximately simulate, predict, and control the system using $N$-dimensional linear methods. These eigenfunctions may be learned from data via the use of SVD on the trajectories, a predetermined dictionary of reference functions, or a neural network, which inspires the classical DMD and eDMD algorithms. The linear coordinates are often learned jointly with the dynamics in this linear representation via regression. In short, this pipeline learns a lifting to a linear system on which system identification is performed. See the survey (Brunton et al., 2021) for more details about this approach.

In contrast, our method is *improper*, i.e. it does not learn the mapping of the nonlinear dynamical system to higher dimensions explicitly. We rely on the spectral filtering method to automatically learn the best predictor that competes with a linear predictor with very high-dimensional hidden state.

## 2.2. Methods for Learning in Linear Systems

The simplest and most commonly studied dynamical systems in the sciences and engineering are linear. Given input vectors $u_1, \ldots, u_T \in \mathbb{R}^{d_{\text{in}}}$, the system generates a sequence of output vectors $y_1, \ldots y_T \in \mathbb{R}^{d_{\text{out}}}$ according to the following equations

$$x_{t+1} = Ax_t + Bu_t + w_t \tag{2.1}$$
$$y_t = Cx_t + Du_t + \xi_t,$$

where $x_0, \ldots, x_T \in \mathbb{R}^{d_h}$ is a sequence of hidden states, $(A, B, C, D)$ are matrices which parameterize the LDS, and $w_t, \xi_t$ are perturbation terms. We assume w.l.o.g. that $D, \xi_t = 0$, as these terms can be folded into the input of the previous iteration. The problem of *prediction* in such systems refers to predicting the next observation $y_t$ given all previous inputs $u_{1:t}$ and past observations $y_{1:t-1}$.

We survey the problem of prediction in linear dynamical systems (LDS) and how our results advance it in Appendix A. To summarize, linear systems become difficult due to either marginal stability (eigenvalues of $A$ near the unit circle, which create long memory), complex system eigenvalues (which create oscillation), or non-normality (which create transient behavior). Despite this, simple improper learning algorithms can learn linear systems well; a state of the art is spectral filtering, which competes against marginally-stable LDS's with real system eigenvalues.

## 3. Setting

We consider nonlinear dynamical systems of the form (1.1), which generate a sequence $y_1, \ldots, y_T$ of observations. Our main results are proved as loss bounds in realizable settings, i.e. when the data is generated from a ground truth dynamical system. Since the derivations involved go through the language of online convex optimization, we expect that these can be converted to agnostic regret guarantees.

**Assumption 3.1** (Nonlinear data generator). The signal $(y_t)_{t=1}^T$ is generated by a nonlinear dynamical system $(f, h, x_0)$ as per equations (1.1). We assume that:

(i) For all $t$, the states and signal are bounded by $\|x_t\|, \|y_t\| \leq R$.

(ii) The dynamics $f$ and observation $h$ are 1-Lipschitz.

### 3.1. Notation

We use $\mathbb{D}_r$ to denote the complex disk of radius $r$, i.e. $\mathbb{D}_r = \{\lambda \in \mathbb{C} : |\lambda| \leq r\}$. We use the notation $v_{a:b}$ to

denote stacking the sequence $v_a, v_{a+1}, \ldots, v_{b-1}, v_b$. We allow indices to be negative for notational convenience, and quantities with negative indices are understood as zero. We define $\log_+(x) = 1 + \max(0, \log(x))$ for positive $x$. Unless otherwise stated, the norm $\|\cdot\|$ refers to the Euclidean $\ell_2$ norm for vectors and the corresponding operator/spectral norm for matrices. $\|\cdot\|_F$ denotes the Frobenius norm of a matrix. For $A \in \mathbb{R}^{d \times d}$ complex diagonalizable we let

$$\kappa_{\mathrm{diag}}(A) = \|H\| \cdot \|H^{-1}\|$$

be the condition number of the best similarity transformation $H \in \mathbb{C}^{d \times d}$ for which $A = HDH^{-1}$ with $D$ diagonal, which is sometimes called the "spectral condition number" of $A$. If $A$ is not diagonalizable we set $\kappa_{\mathrm{diag}}(A) \equiv +\infty$, and if $A$ is normal then $\kappa_{\mathrm{diag}}(A) = 1$. We use $\Pi_{\mathcal{K}}$ to denote nearest-point projection to a convex set $\mathcal{K}$. We use the notation that $\widetilde{O}(\cdot)$ hides logarithmic dependence.

## 4. Algorithm and Main Result

Our algorithm leverages spectral filtering of past outputs to build a predictor. By filtering over past observations $y_{t-1}, y_{t-2}, \ldots$, the algorithm has a way to approach nonlinear dynamics, asymmetric linear dynamics, and adversarial process noise. This is implemented[1] in Algorithm 1.

Henceforth, for simplicity of presentation we assume that the dimension of the observation is one, i.e. $d_{\mathcal{Y}} = d_{\mathrm{obs}} = 1$ (this is the hardest case). The details that change when the observation space is larger are noted in Appendix F.

Theorem 4.1 states the learning guarantee this algorithm achieves for nonlinear dynamical systems. We will use the construction of the Luenberger program, defined precisely in Definition B.8, with an optimal value $Q_\star$ which depends on the dynamical system. In short, $Q_\star$ measures how complicated a linear observer system which approximates the observations must be, where we require the system eigenvalues to be either real (in $[0, 1-\rho] \cup \{1\}$) or strongly stable (in $\mathbb{D}_{1-\gamma}$).

**Theorem 4.1** (Nonlinear). *Let $y_1, \ldots, y_T$ be the observations of a nonlinear dynamical system $(f, h)$ satisfying Assumption 3.1, and fix $0 < \rho \le \gamma < 1$. Let*

$$Q_\star = \limsup_{\varepsilon \downarrow 0} Q_\star\left(f, h, \varepsilon, [0, 1-\rho] \cup \{1\} \cup \mathbb{D}_{1-\gamma}\right)$$

*be as specified in Definition B.8.*

*If $\hat{y}_1, \ldots, \hat{y}_T$ are the predictions made by running Algorithm 1 with $J^t, M^t \equiv 0$ and parameters $h = \Theta(\log^2(Q_\star T))$,*

---

[1]To avoid restating many variations of the same algorithm, we state Algorithm 1 over both open-loop controls $u_t$ and observations $y_t$. It is understood that when applying our results to a nonlinear system such as (1.1), since there are no $u_t$'s we take $J^t, M^t \equiv 0$.

---

**Algorithm 1** Observation Spectral Filtering + Regression

---

1: **Input:** Horizon $T$, number of filters $h$, number of autoregressive components $m$, step sizes $\eta_t$, convex constraint set $\mathcal{K}$, prediction bound $R$.

2: Compute $\{(\sigma_j, \phi_j)\}_{j=1}^k$ the top eigenpairs of the $(T-1)$-dimensional Hankel matrix of (Hazan et al., 2017).

3: Initialize $J_j^0, M_i^0 \in \mathbb{R}^{d_{out} \times d_{in}}$ and $P_j^0, N_i^0 \in \mathbb{R}^{d_{out} \times d_{out}}$ for $j \in \{1, \ldots, m\}$ and $i \in \{1, \ldots, h\}$. Let $\Gamma^t = (J^t, M^t, P^t, N^t)$ for notation.

4: **for** $t = 0, \ldots, T-1$ **do**

5:    Compute

$$\hat{y}_t(\Gamma^t) = \sum_{j=1}^{m-1} J_j^t u_{t-j} + \sum_{i=1}^{h} \sigma_i^{1/4} M_i^t \langle \phi_i, u_{t-2:t-T} \rangle$$
$$+ \sum_{j=1}^{m} P_j^t y_{t-j} + \sum_{i=1}^{h} \sigma_i^{1/4} N_i^t \langle \phi_j, y_{t-1:t-T-1} \rangle,$$

   and if necessary project $\hat{y}_t(\Gamma^t)$ to the ball of radius $R$ in $\mathcal{Y}$.

6:    Compute loss $\ell_t(\Gamma^t) = \|\hat{y}_t(\Gamma^t) - y_t\|^2$.

7:    Update $\Gamma^{t+1} \leftarrow \Pi_{\mathcal{K}}\left[\Gamma^t - \eta_t \nabla_\Gamma \ell_t(\Gamma^t)\right]$.   # can use other optimizer (such as VAW, see Algorithm 2)

8: **end for**

---

$m = \Theta(\log(Q_\star T/\gamma)/\gamma)$*, and the VAW forecaster with* $\lambda = \Theta\left(\frac{1}{Q_\star^{20}(h+\frac{1}{\gamma})}\right)$*, then*

$$\sum_{t=1}^T \|\hat{y}_t - y_t\|^2 \le O\left(R^2 \frac{\log_+(RQ_\star T/\gamma)^3}{\gamma} + R^2 \frac{\log_+(Q_\star)}{\rho}\right).$$

*Remark* 4.2. The value $Q_\star(f, h, \varepsilon, \Sigma)$ measures the non-normality of the simplest observer system (with all eigenvalues in $\Sigma$) for the linearization of $(f, h, x_0)$ obtained by $\varepsilon$-discretizing the state space $\mathcal{X}$ and considering the induced Markov chain. As $\varepsilon \downarrow 0$ this approaches the non-normality of an observer system of the Perron-Frobenius operator (the dual of the Koopman operator), see Remark B.6 for more details.

*Remark* 4.3. We use the Vovk-Azoury-Warmuth (VAW) forecaster for its logarithmic dependence on the diameter of the optimization set, see Appendix C.1 for the precise update step. One can also run Algorithm 1 as stated using vanilla gradient descent – in practice they appear to perform similarly.

For some intuition, in Lemma D.3 we show that $\log Q_\star$ is upper bounded by $\widetilde{O}(N)$, where $N$ is the number of eigenvalues of a linearization which lie outside $[0, 1-\rho] \cup \{1\} \cup \mathbb{D}_{1-\gamma}$ (assuming these eigenvalues are well-spaced and quantitatively observable). We can informally state this result as follows:

**Corollary 4.4** (Informal). *Let $N$ be the number of marginally-stable complex system eigenvalues of an LDS approximation to the original nonlinear system. Then, $\log(Q_\star) = \widetilde{O}(N)$ and so running Algorithm 1 with $\widetilde{O}(N^2)$ parameters achieves loss*

$$\sum_{t=1}^{T} \|\hat{y}_t - y_t\|^2 \leq \widetilde{O}\left(R^2 N^3\right).$$

As we will see through our analysis, the quantity $Q_\star$ yields a natural measure of the complexity for the improper learning of nonlinear dynamical systems with linear predictors, and the notion of "desirable" eigenvalues allows us to specify the analysis to different classes of improper algorithms. The above results are specific to Algorithm 1, which competes well against systems with either real or strongly stable eigenvalues (see Lemma B.4).

In some cases $Q_\star$ can become large, as highlighted by the following example (see also Proposition E.2 for a nonlinear example based on cryptographic assumptions):

*Example* 4.5 (Permutation system). If $(f, h)$ is a permutation system (i.e. $y_t = v_{t \mod d}$ for a fixed sequence $v_1, \ldots, v_d$), then any linear approximation will have $d$-many complex eigenvalues on the unit circle, which implies complexity $\log Q_\star = \widetilde{\Theta}(d)$ (see Example D.5) and therefore that Algorithm 1 learns with a number of parameters polynomial in $N_\star = \Theta(d)$. Note the lower bound of $\Omega(d)$ parameters and runtime, since any algorithm must somehow see and memorize the sequence $v_1, \ldots, v_d$ (see Proposition E.1).

The quantity $Q_\star$ appears to adapt to the hardness of problem; when it is small the algorithm can learn, and it is large in lower bound instances when the algorithm cannot learn. Importantly, there are natural conditions for low complexity, such as detailed balance conditions and phase bounds, which guarantee small $Q_\star$ and demonstrate that Algorithm 1 can efficiently learn many nonlinear systems. We discuss learnability of nonlinear systems and the implications of this complexity measure more in Section 6.

## 5. Proof Overview

Our proof of Theorem 4.1 makes strong use of online convex optimization and improper learning/convex relaxation. There are three main steps: (1) linearization, (2) eigenvalue placement/observer systems, and (3) spectral filtering. We defer the proofs to Appendix B and sketch the argument here.

1. For a marginally-stable observable nonlinear system (i.e. $(f, h)$ satisfying Assumption 3.1), we construct a very high-dimensional LDS which approximates the output sequence of the original nonlinear system. This global linearization is achieved via discretized Markov chains[2] (an approximation to the Perron-Frobenius operator), see Lemma B.5. Via the regret formulation, this reduces the original nonlinear sequence prediction task to competition against high-dimensional linear systems.

2. Given a candidate high-dimensional approximating LDS, we use the Luenberger observer construction and the linear algebraic theory of pole placement to construct an equivalent LDS with desirable spectral structure. The cost of doing so is precisely what is measured by $Q_\star$. In other words, $Q_\star$ is the complexity/non-normality of the least complex high-dimensional linear system with desirable spectral structure which approximates $(f, h)$.

3. In the first two steps, we constructed a high-dimensional LDS with real or stable eigenvalues whose outputs approximate the original system, with $Q_\star$ measuring its complexity. This is only an existence result: it would be unfeasible to try and directly learn this competitor LDS or to implement our construction. Instead, we make use of the framework of online convex optimization (Hazan, 2016) and regret guarantees like Theorem 1 of (Hazan et al., 2017) to make comparable predictions efficiently.

Since the $\varepsilon$-approximate LDS recovers the original dynamics as $\varepsilon \downarrow 0$ (this is where we use the nonexpansive property, see Lemma B.5), this means that the quantity

$$Q_\star = \limsup_{\varepsilon \downarrow 0} Q_\star(f, h, \varepsilon, \Sigma)$$

truly captures the complexity of the data with respect to linear observers (of arbitrarily large hidden dimension) which have spectrum in $\Sigma$. This reduction pairs nicely with the improper learning guarantee for spectral filtering (Theorem A.3), which ensures that Algorithm 1 predicts as well as any linear observer with spectrum in $\Sigma = [0, 1-\rho] \cup \{1\} \cup \mathbb{D}_{1-\gamma}$.

Our proof reduces the nonlinear case to the linear one, which it then reduces to improper learning over a spectrally-constrained class of LDS predictors with cost measured by $Q_\star$. Some adjustments to the original spectral filtering proofs are required, such as the addition of autoregressive components and improving the dependence on initial conditions via a spectral gap.

---

[2]This approach is in line with the recent work of (Liu et al., 2023), which proved that global linearization can often require discontinuity. Such discontinuity is very natural in the Markov chain picture, but can be difficult to capture via Koopman operators.

## 6. Discussion

**Interpretation of loss bounds.**  Our analysis yields a loss bound of the form

$$\sum_{t=1}^{T} \|\hat{y}_t - y_t\|^2 \le O\left( R^2 \cdot \frac{\text{poly} \log(Q_\star, T)}{\rho} \right)$$

for $(y_t)_{t=1}^{T}$ generated by a marginally stable nonexpansive bounded nonlinear system, where we run Algorithm 1 with $O(\text{poly} \log(Q_\star, T))$ parameters and the Vovk-Azoury-Warmuth forecaster as our optimizer. Here, $Q_\star$ measures how much effort is required to observe the sequence by a linear observer with eigenvalues in $[0, 1-\rho] \cup \{1\} \cup \mathbb{D}_{1-\gamma}$. In effect, $1/\rho$ is a burn-in/mixing time for a theoretical linear observer competitor which has $\log Q_\star$ small enough. For systems which have mostly real or strongly stable eigenvalues, this ensures quick decrease of the loss. To better understand this result, Lemma D.3 says that the loss and number of parameters scales polynomially in the number of undesirable eigenvalues (assuming they are separated and uniformly observable).

We arrive at the main takeaway of our learning results: if a nonlinear system has some high-dimensional approximating LDS with a certain spectral structure, then it can be efficiently learned by an improper algorithm suited for that structure. With the choice of Algorithm 1 as our improper learning method, we get the statement that "*the complexity of learning nonlinear signals via Algorithm 1 scales with the minimal number of marginally stable complex eigenvalues over all linear approximations*".

**Spectral properties of linearizations.**  Note that the spectral properties of Koopman operators depend on the choice of function space. Though we do not make this precise, by nature of improper learning our analysis implies that if there exists *any* function space on which the (truncation of the) Koopman operator satisfies the above desiderata then spectral filtering will succeed. The particular choice of linearization technique used for Theorem 4.1 (state space discretization + Markov chain) inherits the spectral properties of the Perron-Frobenius operator (and therefore of the Koopman operator), which provides an avenue for controlling $Q_\star$ using qualitative properties of the nonlinear system. We expect that more careful mesh constructions/linearizations will extend Theorem 4.1's guarantee to more general systems.

As a concrete example, many physical systems $f$ which satisfy some detailed balance condition (e.g. Langevin dynamics, see Section H.4) have a Koopman operator which can be made self-adjoint, which yields $Q_\star(f, h, \varepsilon, [-1, 1]) = O(1)$ for all $\varepsilon$ and $h$. In other words, for many physical systems we truly do expect the existence of a high-dimensional approximating LDS with few marginally stable complex eigenvalues (see Section H.5 for numerics regarding this).

Combining this with the previous conclusion, we get the statement that "*Algorithm 1 can learn physical systems provably and efficiently with few parameters*".

**A control-theoretic perspective on learnability.**  High-dimensional LDS's are in a sense universal dynamical computers, since by e.g. Lemma B.5 any nonexpansive nonlinear system can be approximated to arbitrary accuracy within this class. As such, it is very natural to define "learnability" of a nonlinear signal in terms of learnability of *some* high-dimensional linear approximation.

Armed with these guiding principles, a central theme in our analysis is the use of control-theoretic tools – Luenberger observer systems and global linearization via Koopman-like methods – not as components to be explicitly designed or learned, but to construct a theoretical benchmark in the form of a high-dimensional LDS predictor that an algorithm should aspire to match. By design, $Q_\star(f, h, \Sigma)$ serves as a fundamental measure of the difficulty of learning a system $f$ from its observations $h$ using an improper algorithm that competes against LDS's with spectrum in $\Sigma$.

This yields a notion of learnability that is tailored to a given improper algorithm. For example, spectral filtering naturally competes against real-diagonalizable LDS's ($\Sigma \subset [-1, 1]$, i.e. physical systems with detailed balance) and regression naturally competes against stable LDS's ($\Sigma \subset \mathbb{D}_{1-\gamma}$, i.e. systems with an attractive fixed point), and Algorithm 1 demonstrates that we can compose these building blocks to cover a larger spectral region. The ability of these algorithms to learn a nonlinear signal depends sharply on the corresponding $Q_\star$ values of the generating system via our regret bounds, and our results give a way to determine the natural algorithm for a nonlinear system based on its qualitative properties (such as reversibility or strong stability).

**Prediction is easier than identification.**  Our algorithmic advantages stem from the fact that we only desire accurate predictions and not recovery of any representation of the dynamics. For starters, this enables algorithms with parameters and runtime independent of a linearization's hidden dimension. The more subtle benefit of this approach is that via improper learning we can compete against the best linearization and observer system without explicitly finding it. Our predictor need not commit to a particular invariant subspace of the Koopman operator nor a particular observer gain matrix, both of which are notoriously difficult to find by first-order methods and dangerous to estimate incorrectly (see Remark B.2). This provides extra robustness: we never worry about missing a system mode, as the algorithm's resources are allocated to the modes optimally.

**Summary.**  To summarize in a few words, the Luenberger program provides a new lens through which to analyze dy-

namical systems: by characterizing their $Q_\star$ values, one can make principled a priori judgments about their amenability to learning from partial observations with improper methods. Algorithm 1 is a very good general choice for such a method, since it can efficiently and provably learn to predict systems that are either physical or strongly stable. This methodology is universal, since the algorithm can be run without needing to know anything about the system in advance: one may simply choose some parameters $m, h$ and apply online least squares to yield performance comparable to e.g. the best handcrafted observer system over a truncated Koopman linearization.

# 7. Experiments

In this section, we aim to numerically confirm our theoretical approach. We investigate the algorithm's performance on a particular nonlinear next-token prediction task[3]. In Appendix H we present some more experiments, such as an asymmetric LDS, the double pendulum, and Langevin dynamics. In addition, in Appendix H we provide some numerical evidence that global linearizations of these nonlinear systems have desirable system eigenvalues and therefore small $Q_\star$.

We run four algorithms for next-token prediction. Two of them correspond to learning a linear predictor over the stream of $y_t$'s via either parameterizing the observer LDS (LDS) or spectral filtering as in Algorithm 1 (SF). In addition, the classical eDMD algorithm for Koopman operator learning is directly parameterizing an LDS over a lifted family of observables, and we introduce SFeDMD which learns a spectral filtering predictor over the lifted observables. For details, see Section H.2.

## 7.1. Lorenz System

The Lorenz system is a 3-dimensional deterministic nonlinear dynamical system with state $x = [x^{(1)}, x^{(2)}, x^{(3)}] \in \mathbb{R}^3$ evolving according to the coupled differential equations[4]

$$\begin{cases} \dot{x}^{(1)} = \sigma(x^{(2)} - x^{(1)}), \\ \dot{x}^{(2)} = x^{(1)}(\rho - x^{(3)}) - x^{(2)}, \\ \dot{x}^{(3)} = x^{(1)}x^{(2)} - \beta x^{(3)} \end{cases}$$

for fixed constants $\sigma, \rho, \beta$. This was first considered by Edward Lorenz in 1963 as a simplified model for atmospheric dynamics, and since then it has become a standard dynamical system on which to test prediction algorithms. For the

---

[3]We choose to present the Lorenz system here (despite the fact that it violates Assumption 3.1(ii)) to demonstrate the generality of our approach: we expect that linear improper learning algorithms can predict many nonlinear systems.

[4]We simulate this ODE using a simple explicit Euler discretization with $\Delta t = 0.01$ to generate our plots and datasets.

choices $\sigma = 10, \rho = 28, \beta = \frac{8}{3}$ (which we make from now on) the dynamics become quite complicated, with chaotic solutions and the existence of a strange attractor of fractal dimension.

For each fixed initial condition $x_0$, the Lorenz system produces a sequence of states $x_1, \ldots, x_T$. We consider two prediction tasks: one in which the algorithm sees full observations $y_t = x_t$, and one in which the algorithm sees partial observations via $y_t = Cx_t$ for $C \in \mathbb{R}^{1 \times 3}$. We run the four algorithms from Section H.2 on both the fully and partially observed versions of the task, with initial conditions and the value of $C$ i.i.d. standard Gaussian. Losses are plotted in Figure 1, averaged over random seeds and smoothed.

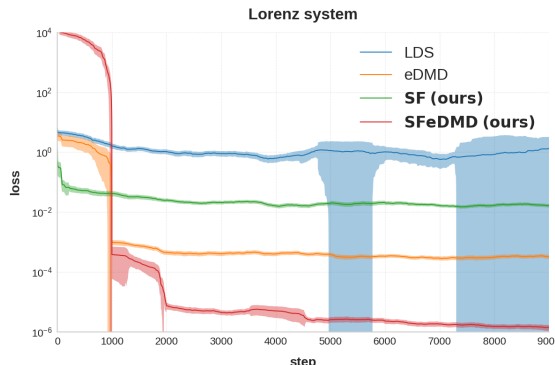

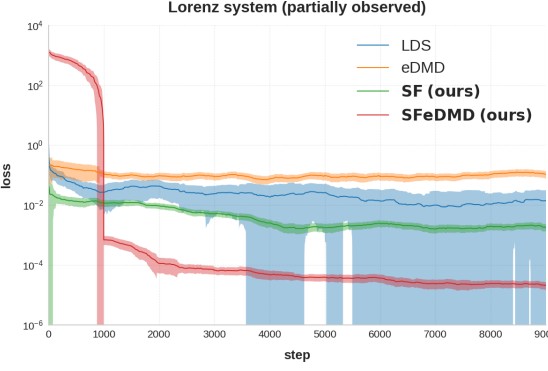

*Figure 1.* Instantaneous losses $\|\hat{y}_t - y_t\|^2$ plotted on log scale as a function of $t$ for the fully and partially observed Lorenz system, respectively, averaged over 12 random seeds with a smoothing filter of length 1000. Shading indicates $\pm 1$ standard deviation.

As we see, Algorithm 1 is able to learn successfully in either the partial or full observation setting when filtering over either the original or nonlinearly-augmented observations – this corroborates the conclusion of Theorem 4.1 even though Assumption 3.1(ii) is violated. By contrast, attempting to learn an observer system directly suffers from instability and nonconvexity, making it unfeasible even on this fairly low-dimensional task. Furthermore, we see that spectral filtering over a sequence augmented by nonlinear observables is certainly the best way to go: the SFeDMD method is able to outperform eDMD with fewer parameters in the

fully observed setting, despite the fact that they both make use of the same dictionary of observables. In addition, a filtering-based predictor is able to aggregate information across time in a way that can handle partial observations naturally, whereas `eDMD` cannot[5].

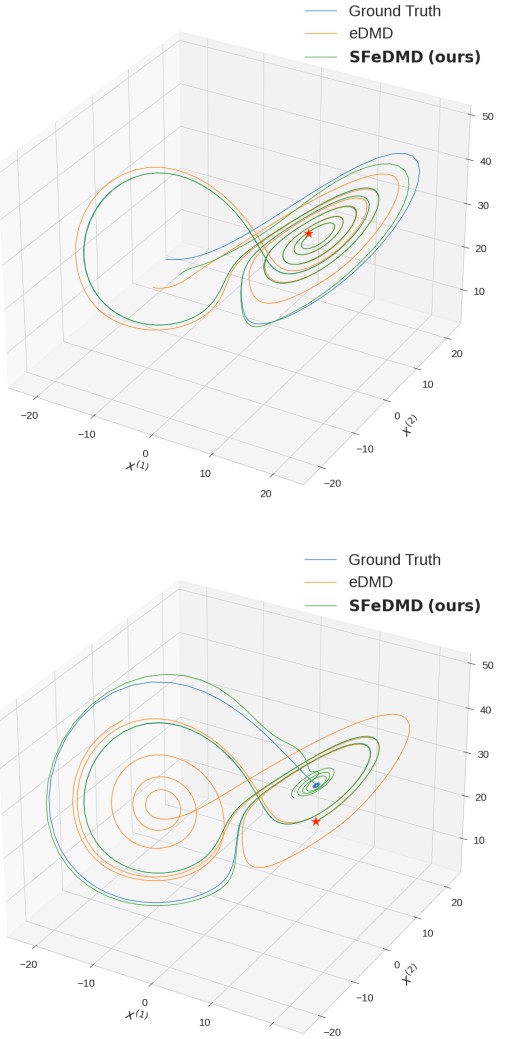

*Figure 2.* Two sets of autoregressive trajectories of length 512, plotted alongside the ground truth trajectory given by continuing to simulate the Lorenz ODE. The initial positions at which the rollouts start are marked with a red star.

So far, we have investigated the performance of various methods on the task of next-token prediction; after all, this is the loss on which we are training and the regret metric that

we use for our theory. While this is by far the most common objective in practice, predictors learned under this loss are often deployed in an autoregressive way[6], such as for text generation in LLMs or rollouts of world models in MPC-type control and planning applications. While our theory does not capture this type of generalization (a predictor which gets very small next-token prediction loss may fail catastrophically in rollout), we find it interesting to inspect the autoregressive behaviors of our learned predictors.

In Figure 2 we plot two autoregressive trajectories of the learned `eDMD` and `SFeDMD` predictors[7] started from contexts unseen during training. On the first one, both methods are able to follow the general trend of the dynamics during the whole rollout, though `SFeDMD`'s predictions remain sharp for longer. On the second trajectory, the errors made by `eDMD` combine with the chaotic nature of this system to produce a qualitatively different rollout, which `SFeDMD` is able to avoid. Of course, the ability of a predictor to generalize from next-token training to autoregressive rollout is heavily dependent on the parameterization, and deep models or different nonlinear liftings likely produce different behaviors. Since autoregressive rollout is not the goal of this paper, we leave a more detailed investigation to future work.

## 8. Conclusions

Deviating from the extensive literature on learning in dynamical systems, we introduced a new approach based on improper learning and spectral filtering. Using classical tools from control theory, including the Koopman operator and the Luenberger observer framework, we established the existence of a real-diagonalizable linear dynamical system that is observationally equivalent to a given nonlinear system. Spectral filtering over this system then enables provable and efficient learning via convex optimization.

Our methodology is accompanied by experiments on both linear and nonlinear systems, validating the predicted efficiency, robustness, and accuracy, and demonstrating regret scaling with the control-theoretic condition number $Q_\star$.

As expected, nonlinear augmentation of observations im-

---

[5]The classical eDMD approach to handle partial observation would be to use time-delayed observables to generate an injective representation of state via Takens' theorem, but (1) anecdotally this is easier said than done since the dynamics of a time-delay lifting are often much more complex and (2) this vastly increases the number of parameters and the complexity of gradient-based optimization.

[6]By autoregression, we mean that there is an initial context $y_1, \ldots, y_L$, which the model uses to produce a prediction $\hat{y}_{L+1}$. In the next iteration, the model uses the context $y_1, \ldots, y_L, \hat{y}_{L+1}$ to predict $\hat{y}_{L+2}$, which in turn gets appended to the context and the process is repeated. In this way, the predictor may be queried with contexts corrupted by its prior mistakes, which allows errors to compound.

[7]The `LDS` and `SF` predictors consistently diverge during rollouts, which may be due to their online training via gradient descent. At each iteration of training the predictor is adapted to the current state and dynamics, and so it may forget about behaviors that it saw earlier in training. For this reason we suspect that training in batch on a dataset of trajectories may fix this.

proves the performance of an otherwise linear predictor. More broadly, we expect much of the two-stage eDMD paradigm—learning or fixing a nonlinear dictionary of observables followed by learning lifted dynamics—to benefit from spectral filtering. Our proposal leaves the lifting step unchanged, and instead replaces direct LDS parametrization with spectral filtering for more efficient prediction of lifted dynamics.

Deep architectures with alternating MLP and SSM layers, which are currently state of the art for learning dynamical systems, can be viewed as jointly learning a nonlinear lifting and linear dynamics in the lifted space.[8] The strong empirical performance of `SFeDMD` over `eDMD` motivates the use of STU (Liu et al., 2024) in place of direct SSM layers, as spectral filtering more effectively exploits nonlinear liftings for sequence prediction—improving accuracy, parameter efficiency, and optimization robustness without sacrificing efficiency.[9] Large-scale experiments on more complex tasks are left to future work.

## 9. Future Work

On the theoretical side, a natural next step is to remove the $\frac{1}{\sqrt{T}}$ spectral-gap requirement in the observer system, which appears to have limited practical impact. While spectral filtering already supports the $\rho = 0$ case (see the remark after Lemma B.4), this would require constructing a high-dimensional LDS approximation with small $R_C$ and $R_{x_0}$. We expect this to be achievable for many nonlinear systems by using richer Koopman operator techniques, which may also allow us to relax Assumption 3.1. Additional extensions include handling adversarial disturbances via reductions to additive LDS noise and incorporating open-loop inputs in nonlinear systems, which remain challenging within both the Koopman and discretized Markov chain frameworks. We leave these directions to future work.

Perhaps the most important next step is to apply the "competition against linear observers" paradigm to the settings of many-step prediction and/or autoregressive rollout. Our current work teaches us the lesson that prediction is easier than system identification and that tools from improper learning/online convex optimization allow us to turn this into an algorithmic advantage: we expect the same story to hold beyond the current setting of single-step prediction.

---

[8]This paradigm is also common in robotics control and planning, where an embedding (e.g., from pixel space) is learned jointly with a latent world model.

[9]At inference time, fast online convolution (Agarwal et al., 2025) or LDS distillation methods (Shah et al., 2025) can recover the same generation efficiency as direct LDS parameterizations.

## Impact Statement

This paper is primarily theoretical, proving that spectral filtering over observations learns to predict some nonlinear systems via competition against linear observers. We believe there are no societal consequences of our work that require specific highlighting here.

## Acknowledgments

Elad Hazan acknowledges support from the Office of Naval Research and Open Philanthropy.

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

# A. Linear Dynamical Systems

Recall that a linear dynamical system (LDS) is parameterized by system matrices $(A, B, C)$. For input vectors $u_1, \ldots, u_T \in \mathbb{R}^{d_{in}}$, the system generates a sequence of output vectors $y_1, \ldots y_T \in \mathbb{R}^{d_{out}}$ according to equations (2.1), i.e. with dynamics

$$x_{t+1} = Ax_t + Bu_t + w_t, \quad y_t = Cx_t,$$

where $x_0, \ldots, x_T \in \mathbb{R}^{d_h}$ is a sequence of hidden states and $w_t, \xi_t$ are perturbation terms. The problem of *learning* in such systems refers to predicting the next observation $y_t$ given all previous inputs $u_{1:t}$ and past observations $y_{1:t-1}$.

Our proofs for nonlinear systems rely on a reduction to LDS and provable improper learning algorithms via online convex optimization. High-dimensional linear systems are universal dynamical predictors – as we will see, most dynamical systems have linear approximations, which is what inspires our construction of $Q_\star$ via the complexity of linear approximations. In this section we survey the state of the art in LDS prediction and describe how our results advance it.

## A.1. Related Work

The fundamental problem of learning in linear dynamical systems has been studied for many decades. In complex systems the hidden dimension can be very large, and it is thus crucial to avoid explicit complexity in this parameter. Other sources of difficulty come from the spectral properties of the dynamics matrix $A$ such as its spectral radius, which determines how close it is to marginal stability. Asymmetry of $A$ and the presence of process noise $w$ can further complicate the situation. We briefly survey the extensive and decades-spanning literature on this topic, focusing on the main approaches or techniques.

**System identification.** System identification refers to the method of recovering $A, B, C$ from the data. This is a hard nonconvex problem that is known to fail if the system is marginally stable, i.e. has eigenvalues that are close to one. Also, recovering the system means time and memory proportional to the hidden dimension, which can be extremely large. (Hardt et al., 2018) shows that the matrices $A, B, C$ of an unknown and partially observed stable linear dynamical system can be learned using first-order optimization methods.

In the statistical noise setting, a natural approach for learning is to perform system identification and then use the identified system for prediction. This approach was taken by (Simchowitz et al., 2018; Simchowitz & Foster, 2020; Sarkar & Rakhlin, 2019). The work of (Ghai et al., 2020) extends identification-based techniques to adversarial noise and marginally stable systems. The work of (Simchowitz et al., 2019) presented a regression procedure for identifying Markov operators under nonstochastic (or even adversarial) noise. This approach has roots in the classical Ho-Kalman system identification procedure (HO & Kálmán, 1966), for which the first non-asymptotic sample complexity, although for stochastic noise, was furnished in (Oymak & Ozay, 2019). Nonstochastic control of unknown, unstable systems without any access to prior information, also called black-box control, was addressed in (Chen & Hazan, 2021).

**Regression a.k.a. convex relaxation.** The (auto) regression method predicts according to $\hat{y}_t = \sum_{i=0}^m M_i u_{t-i}$. The coefficients $M_i$ can be learned using convex regression, and without dependence on the hidden dimension. The downside of this approach is that if the spectral radius of $A$ is $1 - \delta$, it can be seen that $m \approx \frac{1}{\delta}$ terms are needed. The regression method can be further enhanced with "closed loop" components, that regress on prior observations $y_{t-1:1}$. It can be shown using the Cayley-Hamilton theorem that using this method, $d_h$ components are needed, see e.g. (Kozdoba et al., 2019; Hazan et al., 2017).

**Method of moments.** (Hazan et al., 2020) give a method-of-moments system identification procedure as well as the first end-to-end regret result for nonstochastic control starting with unknown system matrices. Extension of these methods to tensors were used in (Bakshi et al., 2023a), and in (Bakshi et al., 2023b) to learn a mixture of linear dynamical systems.

**Kalman filtering.** Kalman Filtering (Kalman, 1960; Anderson & Moore, 1991) involves recovering the state $x_t$ from observations. While Kalman filtering is optimal under specific noise conditions, it generally fails in the presence of marginal stability and adversarial noise. In addition, its complexity depends on the hidden dimension of the system.

**Spectral filtering.** The technique of spectral filtering (Hazan et al., 2017) combines the advantages of many previous methods. It is an efficient method whose complexity does **not** depend on the hidden dimension, and works even for marginally stable linear dynamical systems. Various extensions of the basic technique were proposed for learning and

control (Hazan et al., 2018; Arora et al., 2018; Brahmbhatt et al., 2025), however these do not apply in full generality. The most significant advancement was the recent result of (Marsden & Hazan, 2025), who gave an extension to certain asymmetric LDS, those whose imaginary eigenvalues are bounded from above.

The main technical contribution of this paper, from which we derive the guarantees for learning nonlinear dynamics, is a spectral filtering method which circumvents previous difficulties and obtains the best of all worlds – an efficient method whose parametrization does not depend on the hidden dimension, and allows for marginal stability and asymmetry. Furthermore, our method naturally allows handling of adversarial perturbations in the regret minimization sense. Table 1 below summarizes the state of the art in terms of the fundamental problem of learning linear dynamical systems, and how our result advances it.

| Method | Marginally stable | $d_{\text{hidden}}$-free | asymmetric | adversarial noise |
|---|---|---|---|---|
| Sys-ID | × | × | ✓ | × |
| Regression, open-loop | × | ✓ | ✓ | ✓ |
| Regression, closed-loop | ✓ | × | ✓ | ✓ |
| Spectral filtering (Hazan et al., 2017) | ✓ | ✓ | × | × |
| USP (Marsden & Hazan, 2025) | ✓ | ✓ | ∼ ✓ | × |
| OSF (ours) | ✓ | ✓ | ✓ | ✓ |

*Table 1.* Methods for learning LDS.

## A.2. Improved Spectral Filtering Guarantee

We define the data generator for the realizable case as follows:

**Assumption A.1** (Linear data generator). The input/output signal $(u_t, y_t)_{t=1}^T$ is generated by an LDS $(A, B, C, x_0)$ as per equations (2.1) with arbitrary perturbations $w_1, \ldots, w_T$. We assume that:

(i) $A$ is diagonalizable with all eigenvalues in the unit disk $\mathbb{D}_1$ and[10] $\kappa_{\text{diag}}(A) = 1$.

(ii) The system is bounded, i.e. $\|B\|_F \le R_B$, $\|C\|_F \le R_C$, and $\|x_0\| \le R_{x_0}$.

(iii) For all $t$, the inputs and outputs are bounded by $\|u_t\|, \|y_t\| \le R$, and $\|w_t\| \le W$.

(iv) The system $(A, C)$ is observable.[11]

*Remark* A.2. The observability assumption is without loss of generality: since we are only interested in prediction we may project out the unobservable subspace (the kernel of the observability matrix) to produce an observable LDS without changing the input/output signal. This is known as Kalman's minimal realization, see Chapter 25 of (Dahleh et al., 2011).

Our main technical tool advances the linear case, and is the second step in the argument: a guarantee for improperly learning marginally stable and asymmetric linear dynamical systems with controls and noise. The proof of the following theorem is deferred to Appendix B: it combines eigenvalue placement, Luenberger observer systems, and the complexity $Q_\star$ with the spectral filtering theory for real-diagonalizable systems.

**Theorem A.3** (Linear). *Let $y_1, \ldots, y_T$ be the observations of an LDS driven by inputs $u_1, \ldots, u_T$ and disturbances $w_1, \ldots, w_T$ for which Assumption A.1 is satisfied. Fix $0 < \rho \le \gamma < 1$ and define the Luenberger complexity*

$$
\begin{aligned}
Q_\star &= Q_\star \left( A, C, [0, 1-\rho] \cup \{1\} \cup \mathbb{D}_{1-\gamma} \right) \\
&= \min_{L \in \mathbb{R}^{d_h \times d_{\text{out}}}} \left\{ \kappa_{diag}(A - LC) : \quad A - LC \text{ has all eigenvalues in } [0, 1-\rho] \cup \{1\} \cup \mathbb{D}_{1-\gamma} \right\}.
\end{aligned}
$$

---

[10]For an LDS $(A, B, C)$ with a general diagonalizable $A = HDH^{-1}$, one may consider the equivalent LDS $(D, H^{-1}B, CH)$, in which case the product $R_B R_C$ which appears in our regret bounds scales up by $\kappa_{\text{diag}}(A)$.

[11]This means that the matrix $\mathcal{O} = [C, CA, CA^2, \ldots, CA^{d_h - 1}] \in \mathbb{R}^{d_{\text{obs}} d_h \times d_h}$ has rank $d_h$. In the case of 1-dimensional observations, observability is equivalent to invertibility of $\mathcal{O}$, and in general it describes whether the observation space sees all directions of state space. If $A$ has an eigenvalue with multiplicity $> d_{\text{obs}}$, then $(A, C)$ cannot be observable. See Section 10.1 of (Hazan & Singh, 2025) for more information.

*Denote the quantities*

$$Q = R_C Q_\star \cdot \left( R_B + \frac{2\sqrt{d_\mathcal{Y}} Q_\star}{\sigma_{\min}(C)} \right), \quad R_0 := \max\left\{ R + \frac{R_C Q_\star W}{\rho}, \, 1 \right\}$$

*and let $\hat{y}_1, \ldots, \hat{y}_T$ be the sequence of predictions that Algorithm 1 produces when run with $h = \Theta(\log^2(QT))$, $m = \Theta(\log(QT/\gamma)/\gamma)$, and the VAW forecaster using $\lambda = \Theta\left(\frac{1}{Q^2(h+\frac{1}{\gamma})}\right)$. Then, we have*

$$\sum_{t=1}^{T} \|\hat{y}_t - y_t\|^2 \le O\left( R_0^2 \cdot d_\mathcal{Y}(d_u + d_\mathcal{Y}) \cdot \frac{\log_+(R_0 QT/\gamma)^3}{\gamma} + R_0^2 \cdot \frac{\log_+(R_C Q_\star R_{x_0}/R_0)}{\rho} + \frac{R_C^2 Q_\star^2}{\rho^2} \sum_{t=0}^{T-1} \|w_t\|^2 \right).$$

*Remark* A.4. In the absence of noise this bound scales polylogarithmically with $Q_\star$, whereas the noise term scales polynomially with $Q_\star$. We remark that if the reader is fine with losses scaling polynomially with $Q_\star$ then one may in fact take $h$ and $m$ independent of $Q_\star$ and use any optimizer (such as online gradient descent). Our above choices of $h, m$, and the VAW forecaster are designed to produce the logarithmic scaling for the optimization and approximation terms.

The above theorem provides an adaptive guarantee in the form of a data-dependent loss bound for Algorithm 1. If the system originally had a desirable spectral structure then $Q_\star = O(1)$, in which case the guarantee is strong. By contrast, if the system has very undesirable spectrum then the guarantee is weaker (since $Q_\star$ may be e.g. exponential in the number of undesirable eigenvalues).

We emphasize that the predictions compete with the best choices of $\rho, \gamma$ in hindsight: if there is significant noise then it may be preferable to take larger $\rho$, whereas if there is no noise we can take $\rho$ very small, say on the order of $\frac{1}{\sqrt{T}}$. Without knowing any of these considerations, the algorithm does as well as the best possible arrangement using the resources $m, h$ that it is given. In this way, we predict as well as the best desirable high-dimensional linear observer system for a stream of data, from which Theorem 4.1 will follow.

To summarize, our results for linear dynamical systems exhibit the following important properties:

1. **Asymmetric systems.** The spectral filtering guarantee is extended to capture asymmetric systems, without requiring small or clustered phases of complex eigenvalues as in previous work. The cost for doing so is measured using the non-normality of the best observer system (via $Q_\star$). Of course for very asymmetric systems this cost is high, which is unavoidable (see Proposition E.1 and Example D.5); however, for systems with e.g. few marginally-stable asymmetric modes and/or many almost-real modes the cost is much smaller.

2. **No hidden dimension dependence.** There is no explicit dependence on the hidden dimension $d_h$ in the number of parameters, runtime, or loss. Thus, Algorithm 1 can predict as well as some very high-dimensional linear systems with much less work.

3. **Noise robustness.** Previous results in spectral filtering starting from (Hazan et al., 2017) exhibit compounding loss from noise (or have dependence on hidden dimension, such as Theorem 2 of (Hazan et al., 2018)). This means that in the presence of adversarial noise, the instantaneous prediction error can be as large as $\|\hat{y}_T - y_T\| \approx T\|w\|$. In contrast, Algorithm 1 has $\|\hat{y}_T - y_T\| \lesssim \|w\| + \frac{1}{\sqrt{T}}$. Note that there is a lower bound of $\|\hat{y}_T - y_T\| \gtrsim \|w\|$ for observable noise, see Proposition E.1.

## B. Analysis

In this section we perform our main analysis. We first detail our construction of the Luenberger program and $Q_\star$ and its implications for learning linear systems, culminating in a proof of Theorem A.3. We then describe a reduction from nonexpansive nonlinear systems to high-dimensional linear systems via discretization, from which Theorem 4.1 follows.

### B.1. Regret

The prediction notion we use is borrowed from the setting of online convex optimization (Hazan, 2016), where the performance of an algorithm is measured by its regret. Regret is a robust performance measure, as it does not assume a

specific generative or noise model. It is defined as the difference between the loss incurred by our predictor and that of the best predictor in hindsight belonging to a certain class, i.e.

$$\text{Regret}_T(\mathcal{A}, \Pi) = \sum_{t=1}^{T} \ell(y_t, \hat{y}_t^{\mathcal{A}}) - \min_{\pi \in \Pi} \sum_{t=1}^{T} \ell(y_t, \hat{y}_t^{\pi}).$$

Here $\mathcal{A}$ refers to our predictor, $\hat{y}_t^{\mathcal{A}}$ is its prediction at time $t$, and $\ell$ is a loss function. Further, $\pi$ is a predictor from the class $\Pi$, for example all predictions that correspond to noiseless linear dynamical systems of the form $\hat{y}_{t+1}^{\pi} = \sum_{i=0}^{t} CA^i Bu_{t-i}$. A stronger predictor class is that of all closed-loop linear dynamical predictors that also take into account the initial state and noise, even though they are not observed by an algorithm, i.e.

$$\hat{y}_{t+1}^{\pi} = CA^{t+1}x_0 + \sum_{i=0}^{t-1} CA^i(Bu_{t-i} + w_{t-i}).$$

## B.2. Linear Case

In this subsection, we prove Theorem A.3. Given a linear dynamical system (LDS), we construct another LDS that incorporates past observations as feedback, has a desirable transition matrix, and generates the same trajectories as the original system. This construction relies on the **Luenberger observer** framework (Luenberger, 1971), which we sketch next.

For a given LDS $(A, B, C, x_0)$, we can always choose a gain $L \in \mathbb{R}^{d_h \times d_{\text{out}}}$ and construct the observer system

$$\tilde{x}_{t+1} = A\tilde{x}_t + Bu_t + L(y_t - \tilde{y}_t), \quad \tilde{y}_t = C\tilde{x}_t, \quad \tilde{x}_0 = x_0.$$

This system has the same dynamics as the original one, but with an additional term that uses the observation error $y_t - \tilde{y}_t$ as linear feedback. Since $y_t = Cx_t$ and $\tilde{y}_t = C\tilde{x}_t$, if we define the state error $\xi_t := x_t - \tilde{x}_t$ then we get

$$\begin{aligned}
\xi_{t+1} &= Ax_t + Bu_t + w_t - A\tilde{x}_t - Bu_t - LC(x_t - \tilde{x}_t) \\
&= (A - LC)\xi_t + w_t.
\end{aligned}$$

Rolling this out reveals that for all $t$,

$$x_t = \tilde{x}_t + (A - LC)^t(x_0 - \tilde{x}_0) + \sum_{t=1}^{t} (A - LC)^{i-1} w_{t-i}.$$

It is a known fact in control theory that if $(A, C)$ is observable then we can choose $L$ to place the eigenvalues of $A - LC$ as desired, which we describe further in Appendix D. This is a standard construction which is frequently used with system identification for prediction and control from partial observations (Shastri et al., 2023), and the Kalman filter is related to a special case in which the gain $L$ is chosen optimally under Gaussian noise assumptions (Kalman, 1960). Our approach will be to predict via spectral filtering, which learns improperly and competes in hindsight against all observer systems.

For certain systems $(A, C)$ it is harder to infer hidden state information from observations, which is reflected in terms of the size of $L$ and the non-normality of the observer dynamics $A - LC$. Many Luenberger observer-based methods which require $L$ explicitly end up approaching this through heuristics or structural assumptions (Peralez & Nadri, 2021; Brivadis et al., 2019) or by performing a numerically-difficult optimization over $L$ (Pandey et al., 2014). However, since spectral filtering matches the performance of the *best* observer system in hindsight, we will make use of a quantity that captures the complexity of the best observer system:

**Definition B.1** (Luenberger program). Let $A \in \mathbb{R}^{d_h \times d_h}$ and $C \in \mathbb{R}^{d_{\text{out}} \times d_h}$. For each spectral constraint set $\Sigma \subset \mathbb{C}$ we consider the optimization problem over $L \in \mathbb{R}^{d_h \times d_{\text{out}}}$ given by

$$\min \quad \kappa_{\text{diag}}(A - LC) \quad \text{s.t.} \quad A - LC \text{ has all eigenvalues in } \Sigma.$$

When the constraint set of this optimization is nonempty, denote the minimal value by $Q_\star = Q(A, C, \Sigma) < \infty$ and a minimizing gain by $L_\star$.

We will be applying Algorithm 1 to improperly learn over the class of observer systems. As we will see in Lemma B.4, spectral filtering does well for $\Sigma \subset [0,1]$ and very well for[12] $\Sigma \subset [0, 1-\rho] \cup \{1\}$, and the regression terms are able to handle $\Sigma \subset \mathbb{D}_{1-\gamma}$; so, we will be mostly interested in $\Sigma = [0, 1-\rho] \cup \{1\} \cup \mathbb{D}_{1-\gamma}$. For generality, we will state things for arbitrary $\Sigma$ and specify where needed.

In Appendix D we will say more about this optimization and apply known bounds on $Q_\star$, and in particular we show in Lemma D.3 that for quantitatively observable systems $\log Q_\star \lesssim O(N)$, where $N$ is the number of system eigenvalues outside the desired region $\Sigma$. This can be tight, see Example D.5.

*Remark* B.2. The objective function $Q_\star$ is related to the quantity $\mathcal{S}_2$ from (Mehrmann & Xu, 1998), which describes the stability of eigenvalue placement. For this reason, such minimizations are referred to as "robust pole placement", see also the survey (Pandey et al., 2014). It is known that the eigenvalue placement problem is very ill-conditioned, and small perturbations to the data $(A, C)$ yield $Q_\star$-sized changes in $L$ and the placed eigenvalues (see e.g. Theorem 3.1 of (Mehrmann & Xu, 1996)). If $Q_\star$ is large, then a pipeline based on eigenvalue placement combined with system identification from data can be very brittle.

In words, the value of $Q_\star$ describes how hard it is (measured in terms of conditioning of the closed-loop dynamics) to approximately recover the state using a desirable observer system.[13] We can describe the observer system construction in terms of $Q_\star$:

**Definition B.3** (Observer system). Consider the following LDS

$$x_{t+1} = Ax_t + Bu_t + w_t, \quad y_t = Cx_t.$$

Fix $\Sigma \subset \mathbb{D}_{1-\delta}$ for some $\delta \in [0,1]$, let $Q_\star = Q(A, C, \Sigma)$ be the value of the corresponding Luenberger program from Definition B.1, and suppose that $Q_\star < \infty$. Then, there exists an observer system which takes as inputs $(u_0, \ldots, u_t)$ and $(y_0, \ldots, y_t)$ and produces the estimated observation $\tilde{y}_{t+1}$ via the LDS

$$\tilde{x}_{t+1} = \tilde{A}\tilde{x}_t + Bu_t + Ly_t, \quad \tilde{y}_t = C\tilde{x}_t, \quad \tilde{x}_0 = x_0$$

with the properties:

(i) $\|L\|_F \leq \frac{2Q_\star}{\sigma_{\min}(C)}$, where the denominator is the minimal singular value.

(ii) $\tilde{A} = HDH^{-1}$ where $D$ is a diagonal matrix with entries in $\Sigma$ and $\kappa_{\mathrm{diag}}(\tilde{A}) \equiv \|H\| \, \|H^{-1}\| = Q_\star$.

(iii) $\sum_{t=1}^{T} \|y_t - \tilde{y}_t\| \leq \frac{Q_\star \|C\|}{\delta} \left( \sum_{t=0}^{T-1} \|w_t\| \right)$, and if $w_t \equiv 0$ then $y_t = \tilde{y}_t$ even if $\delta = 0$.

Let $L_\star = L(A, C, \Sigma) \in \mathbb{R}^{d_h \times d_{\mathrm{out}}}$ be a solution of the Luenberger program. Properties (i) and (ii) follow from the objective function and constraint set of the Luenberger program, along with Lemma D.2. As mentioned previously, if we define the errors $\xi_t := x_t - \tilde{x}_t$ then $\xi_t = \sum_{i=1}^{t} (A - L_\star C)^{i-1} w_{t-i}$ since $\xi_0 = 0$; so, in the noiseless setting we exactly reproduce the states and observations. In the noisy case, since $y_t - \tilde{y}_t = C\xi_t$ and all eigenvalues of $\tilde{A} = A - L_\star C$ have magnitude $\leq 1 - \delta$,

$$\sum_{t=1}^{T} \|\tilde{y}_t - y_t\| \leq \|C\| \, Q_\star \sum_{t=1}^{T} \sum_{i=1}^{t} (1-\delta)^{i-1} \|w_{t-i}\| \leq \frac{\|C\| \, Q_\star}{\delta} \cdot \left( \sum_{t=0}^{T-1} \|w_t\| \right)$$

The observer system construction shows that there is an LDS over $(u, y)$ with desirable eigenstructure that nearly matches the original one, which may be rolled out without requiring $w$ (without noise, the observer system matches it exactly). We will combine this with spectral filtering's regret guarantee. In the application of the below lemma, it should be imagined that the inputs $u_t$ are actually the stacked $[u_t, y_{t-1}]$ and we are running the algorithm with $P^t, N^t \equiv 0$ in order to compete against an observer LDS.

---

[12]Negative eigenvalues can be handled by spectral filtering easily with a constant amount of extra parameters and computation, see the extension in Theorem 3.1 of (Agarwal et al., 2023). For ease of presentation, we will continue to state things for nonnegative real eigenvalues.

[13]Other improper algorithms will require a definition of $Q_\star$ tailored to their structure. To maintain generality we will often use the phrases "desirable spectral structure" or "undesirable eigenvalues", where the implied spectral constraint set $\Sigma$ will depend on the algorithm.

**Lemma B.4.** *On any sequence $(u_t, y_t)_{t=1}^T$ such that $\|u_t\|, \|y_t\| \leq R$, Algorithm 1 run with $P_1^t = 1$ and $P_j^t, N^t \equiv 0$ for $j > 1$ using $h = \Theta(\log^2(R_B R_C \kappa T))$, $m = \Theta\left(\frac{1}{\gamma} \log\left(\frac{R_B R_C \kappa T}{\gamma}\right)\right)$, and the VAW forecaster with $\lambda = \Theta(\frac{1}{R_B^2 R_C^2 \kappa^2 (h + \frac{1}{\gamma})})$ achieves regret*

$$\sum_{t=1}^T \|\hat{y}_t - y_t\|^2 - \min_{\pi \in \Pi} \sum_{t=1}^T \|\hat{y}_t^\pi - \hat{y}_{t-1}^\pi - y_t + y_{t-1}\|^2 \leq O\left(R^2 d_u d_y (m + h) \log_+\left(\frac{R R_B R_C \kappa T}{\gamma}\right) + \frac{R^2 \log_+(R_C \kappa R_{x_0}/R)}{\rho}\right),$$

*where $\Pi = \mathrm{LDS}(\rho, \gamma, \kappa, R_B, R_C, R_{x_0}, R)$ is the class of LDS predictors $\pi$ parameterized by $(A, B, C, x_0)$ where $A$ is diagonalizable with eigenvalues in $[0, 1 - \rho] \cup \{1\} \cup \mathbb{D}_{1-\gamma}$ with $\rho \leq \gamma$, $\kappa_{diag}(A) \leq \kappa$, $\|B\|_F \leq R_B$, $\|C\|_F \leq R_C$, $\|x_0\| \leq R_{x_0}$, and $\|\hat{y}_t^\pi\| \leq R$.*

*Proof.* See Appendix C. We remark here that the online least squares is performed on the derivative of the target sequence and the derivative of the predictions made by a competitor. Of course, there is no difference in the realizable setting where $y_t = \hat{y}_t^\pi$ for some $\pi \in \Pi$. $\qquad\square$

*Proof of Theorem A.3.* Let $\Sigma = [0, 1 - \rho] \cup \{1\} \cup \mathbb{D}_{1-\gamma}$. By Lemma D.1, choose $L = L_\star$ such that $\widetilde{A} = A - LC$ has spectrum in $\Sigma$ and $\kappa_{\mathrm{diag}}(\widetilde{A}) \leq Q_\star$. The observer system is $\pi = (\widetilde{A}, [B, L], C, x_0)$ given by

$$\widetilde{x}_t = \widetilde{A}\widetilde{x}_{t-1} + Bu_{t-1} + Ly_{t-1}, \qquad \hat{y}_t^\pi = C\widetilde{x}_t.$$

Equivalently, it is an LDS driven by the lagged augmented inputs $v_{t-1} := (u_{t-1}, y_{t-1})$, with input dimension $d_u + d_y$ and input matrix $\widetilde{B} = [B \; L]$. By Lemma D.2, $\|L\| \leq 2Q_\star/\sigma_{\min}(C)$, so

$$\|\widetilde{B}\|_F \leq \|B\|_F + \|L\|_F \leq R_B + \frac{2\sqrt{d_y} Q_\star}{\sigma_{\min}(C)}.$$

Therefore

$$\|C\|_F \|\widetilde{B}\|_F \kappa_{\mathrm{diag}}(\widetilde{A}) \leq R_C \left(R_B + \frac{2\sqrt{d_y} Q_\star}{\sigma_{\min}(C)}\right) Q_\star = Q.$$

Let $\hat{y}_t^{\mathrm{obs}}$ denote the spectral filtering approximant induced by this observer LDS under the predictor class of Lemma B.4, i.e. it predicts

$$\hat{y}_t^{\mathrm{obs}} = y_{t-1} + (\hat{y}_t^\pi - \hat{y}_{t-1}^\pi).$$

Let $e_t = x_t - \widetilde{x}_t$. Since $y_t = Cx_t$ and $\widetilde{A} = A - LC$,

$$e_t = \widetilde{A}e_{t-1} + w_{t-1}, \qquad e_0 = 0.$$

Thus

$$\hat{y}_t^{\mathrm{obs}} - y_t = y_{t-1} + (\hat{y}_t^\pi - \hat{y}_{t-1}^\pi) - y_t = -C(e_t - e_{t-1}).$$

Expanding $e_t$ gives

$$\hat{y}_t^{\mathrm{obs}} - y_t = -Cw_{t-1} - \sum_{s=0}^{t-2} C\left(\widetilde{A}^{t-1-s} - \widetilde{A}^{t-2-s}\right) w_s.$$

Hence the comparator error is a convolution with kernels $K_0 = C$ and $K_j = C(\widetilde{A}^j - \widetilde{A}^{j-1})$ for $j \geq 1$. Writing $\widetilde{A} = HDH^{-1}$, the unit eigenspace contributes zero to $D^j - D^{j-1}$, while every non-unit eigenvalue has modulus at most $1 - \rho$. Therefore

$$\|K_0\|_{\mathrm{op}} \leq R_C, \qquad \|K_j\|_{\mathrm{op}} \leq 2R_C Q_\star (1 - \rho)^{j-1} \quad (j \geq 1),$$

and so

$$\sum_{j \geq 0} \|K_j\|_{\mathrm{op}} \lesssim \frac{R_C Q_\star}{\rho}.$$

Since $\|w_t\| \leq W$, this also implies $\|\widehat{y}_t^{\mathrm{obs}} - y_t\| \lesssim R_C Q_\star W / \rho$, and therefore

$$\|\widehat{y}_t^{\mathrm{obs}}\| \leq \|y_t\| + \|\widehat{y}_t^{\mathrm{obs}} - y_t\| \lesssim R + \frac{R_C Q_\star W}{\rho} = R_0.$$

Also $\|v_t\| = \|(u_t, y_t)\| \leq \sqrt{2}R \leq O(R_0)$ and $\|y_t\| \leq R \leq R_0$. Thus Lemma B.4 applies to the sequence $(v_t, y_t)_{t=1}^T$ with radius $O(R_0)$, input dimension $d_u + d_y$, condition number $Q_\star$, and product $R_C \|\widetilde{B}\|_F Q_\star \leq Q$. Hence

$$\sum_{t=1}^T \|\widehat{y}_t - y_t\|^2 \leq \sum_{t=1}^T \|\widehat{y}_t^{\mathrm{obs}} - y_t\|^2 + O\left(R_0^2 d_y (d_u + d_y)(m+h)\log_+\left(\frac{R_0 QT}{\gamma}\right) + \frac{R_0^2 \log_+(R_C Q_\star R_{x_0}/R_0)}{\rho}\right).$$

By Young's convolution inequality,

$$\sum_{t=1}^T \|\widehat{y}_t^{\mathrm{obs}} - y_t\|^2 \lesssim \left(\sum_{j \geq 0} \|K_j\|_{\mathrm{op}}\right)^2 \sum_{t=0}^{T-1} \|w_t\|^2 \lesssim \frac{R_C^2 Q_\star^2}{\rho^2} \sum_{t=0}^{T-1} \|w_t\|^2.$$

Combining the last two displays gives

$$\sum_{t=1}^T \|\widehat{y}_t - y_t\|^2 \leq O\left(R_0^2 d_y (d_u + d_y)(m+h)\log_+\left(\frac{R_0 QT}{\gamma}\right) + \frac{R_0^2 \log_+(R_C Q_\star R_{x_0}/R_0)}{\rho}\right) + O\left(\frac{R_C^2 Q_\star^2}{\rho^2}\sum_{t=0}^{T-1}\|w_t\|^2\right).$$

Finally, with $h = \Theta(\log^2(QT))$ and $m = \Theta\left(\frac{1}{\gamma}\log\left(\frac{QT}{\gamma}\right)\right)$ we have

$$(m+h)\log_+\left(\frac{R_0 QT}{\gamma}\right) \leq O\left(\frac{\log_+^3(R_0 QT/\gamma)}{\gamma}\right),$$

which proves the claimed bound. $\qquad\square$

## B.3. Nonlinear Case

In this subsection, we will prove Theorem 4.1. Our main tool is a space-discretized Markov chain argument (related to Ulam's method) that achieves the same global linearization goal as the Koopman operator, which may be of independent interest. The discretization argument is sufficient for sequence prediction and avoids traditional compactness and spectral assumptions, replacing them by simpler boundedness and Lipschitz conditions on the dynamics.

**Lemma B.5.** *Let $y_1, \ldots, y_T$ be generated by a deterministic discrete-time dynamical system $(f, h, x_0)$ satisfying Assumption 3.1. Let $\mathcal{S}$ be an $\varepsilon$-net of the ball of radius $R$ in $\mathcal{X}$. The cardinality of $\mathcal{S}$ satisfies*

$$N = |\mathcal{S}| \leq \left(\frac{2R}{\varepsilon}\right)^{d_{\mathcal{X}}}.$$

*Then, there exists an observable LDS of hidden dimension $\leq N$ with states $z_t \in \mathbb{R}^N$ and an observation matrix $C' \in \mathbb{R}^{d_y \times N}$ with Frobenius norm $\|C'\|_F \leq R\sqrt{N d_y}$, such that the output sequence $y_t' = C' z_t$ satisfies*

$$\sum_{t \leq T} \|y_t' - y_t\| \leq \frac{T^2}{2}\varepsilon.$$

*Proof.* The $\varepsilon$-net condition guarantees that for every $x \in \mathcal{X} \cap B_R$ there exists $s \in \mathcal{S}$ such that $\|x - s\| \leq \varepsilon$. It is standard that such an $\varepsilon$-net can be chosen with $|\mathcal{S}| = N$, and we enumerate the elements as $s^{(1)}, \ldots, s^{(N)}$.

Define the mapping $\pi : \mathcal{X} \to \mathcal{S}$ as the nearest neighbor projection, i.e. $\pi(x) = \mathrm{argmin}_{s \in \mathcal{S}} \|x - s\|$. We consider the deterministic transition operator $T : \mathcal{S} \to \mathcal{S}$ as $T(s) = \pi(f(s))$, yielding a Markov chain with states $s_t$ in a finite state space. If we let $z_t \in \mathbb{R}^N$ be the indicator vectors corresponding to this state and the evolution operator $A' \in \mathbb{R}^{N \times N}$ with $A'_{ij} = 1$ if $T(s^{(j)}) = s^{(i)}$ and 0 else, then we get a linear evolution

$$z_{t+1} = A' z_t, \quad y_t' = C z_t$$

with $C \in \mathbb{R}^{d_{\mathcal{Y}} \times N}$ sending the one-hot vector $z$ corresponding to $s^{(j)}$ to $h(s^{(j)})$. Since $\|y_t\| \leq R$, the matrix elements of $C'$ can be taken to be bounded by $R$.

With these definitions in place, let $x_0, \ldots, x_T$ be a trajectory of the original nonlinear system with observations $y_0, \ldots, y_T$. We let $s_0 = \pi(x_0)$ and $s_{t+1} = T(s_t)$ be the rollout of the finite state space Markov chain, and define $\xi_t = \|s_t - x_t\|$ the error against the true states. Then,

$$
\begin{aligned}
\xi_{t+1} &= \|s_{t+1} - x_{t+1}\| \\
&= \|\pi(f(s_t)) - f(x_t)\| \\
&\leq \|f(s_t) - f(x_t)\| + \|f(s_t) - \pi(f(s_t))\| \\
&\leq \xi_t + \varepsilon,
\end{aligned}
$$

where we used 1-Lipschitzness of $f$ and the $\varepsilon$-net condition in the last line. Since $\xi_0 \leq \varepsilon$ again by the $\varepsilon$-net construction, we find that $\xi_t \leq t\varepsilon$ and so $\|y_t' - y_t\| \leq t\varepsilon$ by Lipschitzness of $h$. Lastly, we may decompose the hidden states into the observable and unobservable parts of $(A', C')$ by diagonalizing (an $\varepsilon$-perturbation of) the observability matrix $\mathcal{O}$. The unobservable parts do not contribute to the output sequence by definition, and so projecting to the observable subspace yields an observable LDS. Note that doing so cannot increase the Frobenius norm of $C'$. $\qquad\square$

This essentially says that a bounded and marginally stable nonlinear dynamical system can be approximated via a $T^{O(d_{\mathcal{X}})}$-dimensional LDS over a horizon of length $T$. An alternate approach to approximating nonlinear systems via high-dimensional linear ones is to use a spectral truncation of the Koopman operator, as outlined in e.g. (Brunton et al., 2021). However, proving convergence of the finite-dimensional spectral truncations of Koopman operators can require strong assumptions or compactification tricks such as different input and output spaces (Powell et al., 2023), without which it becomes difficult to describe convergence of continuous and residual spectra. Furthermore, the mechanism of pole placement via observer systems becomes obscured in such settings.

We adopt the improper learning perspective that the only thing that matters is approximately correct observations $y_t' \approx y_t$, and that convergence of spectra is not strictly necessary for sequence prediction. The discretization and Markov chain outlined above achieves this, and the proof is clean and simple with fewer assumptions.[14]

*Remark* B.6. This Markov chain is closely related to the Perron-Frobenius operator $P$, the adjoint to the Koopman operator. Informally, $P$ is a linear operator on spaces of measures on the state space $\mathcal{X}$ which sends a measure $\mu$ to its pushforward under the dynamics, i.e. $P\mu = f_{\#}\mu$. One could imagine an infinite-dimensional LDS of the form

$$
\mu_{t+1} = P\mu_t, \quad y_t = \int_{\mathcal{X}} h(x) d\mu_t(x), \quad \mu_0 = \delta_{x_0}
$$

which generates the same observations as the original system, where both the dynamics and observations are linear with respect to the evolving measure. The Markov chain in Lemma B.5 is constructing exactly this LDS, but on a discretized state space to ensure finite-dimensionality (i.e. probability measures are represented via probability vectors). This discretization argument circumvents any need for proofs of convergence of spectral truncations of the Koopman operator or its adjoint $P$.

Importantly, the observation that the Markov chain constructed in Lemma B.5 is (dual to) the Koopman operator implies that the discretized LDS retains some spectral properties of the Koopman operator. Technically, since we use the Euclidean norm for the analysis of this discretized LDS we inherit the spectral properties of the Koopman operator on $L^2(\mathcal{X}, \pi)$ function spaces, where the choice of measure $\pi$ is equivalent to the choice of discretization grid. A very useful result of our improper learning formulation of the problem is that our analysis adapts in hindsight to the best choice of discretization grid, i.e. the Koopman operator on a function space that gives it the most desirable spectral properties. In this way, we sidestep the suboptimalities and difficulties of any explicit choices of function space. For the results we state in this paper we stick to the Euclidean setting, and leave other metrics and meshes to future work.

*Remark* B.7. We note here that the Markov chain discretization approach to global linearization is not only simpler than Koopman truncations, but it is sometimes more faithful to the original nonlinear dynamics. In (Liu et al., 2023) it is shown

---

[14]Note, however, that any construction made via truncated Koopman operators will also fall into the comparator class of high-dimensional LDS, against which we have a regret guarantee from Lemma B.4 in the spirit of Theorem A.3. If there were a better high-dimensional LDS that approximates the sequence $y_1, \ldots, y_T$, then Algorithm 1 would match it without needing to know anything about it.

that global linearization often requires discontinuity to capture nontrivial limit sets; this is very natural to represent in a finite state space Markov chain, but can require complicated and opaque constructions on the Koopman operator side.

In any case, we are left with a high-dimensional LDS which approximates the original system, and we will conclude by applying Theorem A.3. First, we define the relevant quantities which we use in the statement and proof of Theorem 4.1.

**Definition B.8** (Nonlinear Luenberger program). Let $(f, h, x_0)$ be a nonlinear dynamical system as per (1.1) with $\|x_t\|, \|y_t\| \leq R$ and $f, h$ being 1-Lipschitz. For any discretization $\varepsilon > 0$ and spectral constraint set $\Sigma \subset \mathbb{C}$, we consider the optimization program over $L \in \mathbb{R}^{\left(\frac{2R}{\varepsilon}\right)^{d_{\mathcal{X}}} \times d_{\mathcal{Y}}}$ given by

$$\min \quad \kappa_{\text{diag}}(A' - LC') \quad \text{s.t.} \quad A' - LC' \text{ has all eigenvalues in } \Sigma,$$

where $A'$ and $C'$ are the Markov chain transition and observation matrices constructed in Lemma B.5. When the constraint set of the optimization is nonempty, with an overloading of notation we denote the minimal value by $Q_\star(f, h, \varepsilon, \Sigma) = Q_\star(A', C', \Sigma) < \infty$ and a minimizing gain by $L_\star$.

*Proof of Theorem 4.1.* Let $(f, h, x_0)$ be a nonlinear dynamical system satisfying Assumption 3.1, and let

$$z_{t+1} = A' z_t, \quad y'_t = C' z_t$$

be the high-dimensional LDS constructed in Lemma B.5 at some scale $\varepsilon$, where $z_0$ is the indicator vector of $\pi(x_0)$. We construct an alternate LDS $(\bar{A}, \bar{C}, \bar{z}_0)$ via

$$\bar{A} := \begin{bmatrix} A' & 0 \\ C'A' - \beta C' & \beta I \end{bmatrix}, \quad \bar{C} := \begin{bmatrix} 0 & I \end{bmatrix}$$

for some leak parameter $\beta \in \mathbb{D}_1$ to be chosen. We know

$$\begin{bmatrix} z_{t+1} \\ y'_{t+1} \end{bmatrix} = \bar{A} \begin{bmatrix} z_t \\ y'_t \end{bmatrix}$$

since $(C'A' - \beta C')z_t + \beta y'_t = C'A' z_t = y'_{t+1}$, and so this still represents the same output trajectory as $(A', C', z_0)$. Now, let $L'_\star$ be a minimizing Luenberger gain for $Q_\star(A', C', \Sigma)$ and define

$$\bar{L} := \begin{bmatrix} L'_\star \\ C'L'_\star \end{bmatrix}.$$

Then, we know that

$$\bar{A} - \bar{L}\bar{C} = \begin{bmatrix} A & -L'_\star \\ C'A - \beta C' & \beta I - C'L'_\star \end{bmatrix},$$

and if we use the change of variables $S = \begin{bmatrix} I & 0 \\ C' & I \end{bmatrix}$ and $S^{-1} = \begin{bmatrix} I & 0 \\ -C' & I \end{bmatrix}$ then we find

$$S^{-1}(\bar{A} - \bar{L}\bar{C})S = \begin{bmatrix} A' - L'_\star C' & -L'_\star \\ 0 & \beta I \end{bmatrix}$$

which is lower-triangular and so

$$\sigma(\bar{A} - \bar{L}\bar{C}) = \sigma(A' - L'_\star C') \cup \{\beta\}.$$

Since $L'_\star$ was chosen to be admissible, this means that if $\beta \in \Sigma$ then $\bar{L}$ is a valid gain which produces desirable spectrum. Now, we choose $\beta$ such that $\text{dist}(\beta, \sigma(A' - L'_\star C')) \geq \Omega(1)$ and denote $X := (A' - L'_\star C' - \beta I)^{-1} L'_\star$. Thus,

$$\begin{bmatrix} A' - L'_\star C' & -L'_\star \\ 0 & \beta I \end{bmatrix} = \begin{bmatrix} I & X \\ 0 & I \end{bmatrix} \begin{bmatrix} A' - L'_\star C' & 0 \\ 0 & \beta I \end{bmatrix} \begin{bmatrix} I & X \\ 0 & I \end{bmatrix}^{-1}.$$

So, if $A' - L'_\star C' = HD'H^{-1}$ for some diagonalizer $H$ then we may diagonalize $\bar{A} - \bar{L}\bar{C}$ using

$$\bar{H} = S \begin{bmatrix} I & X \\ 0 & I \end{bmatrix} \begin{bmatrix} H & 0 \\ 0 & I \end{bmatrix},$$

which implies that

$$\kappa(\bar{H}) \leq (1 + \|C'\|)^2 \cdot \left(1 + \|(A' - L'_\star C' - \beta I)^{-1} L'_\star\|\right)^2 \cdot \kappa(H)$$

$$\leq (1 + \|C'\|)^2 \cdot \left(1 + \frac{\kappa(H)\|L'_\star\|}{\text{dist}(\beta, \sigma(A' - L'_\star C'))}\right)^2 \cdot \kappa(H)$$

$$\lesssim \left(1 + \frac{\sigma_{\max}(C')}{\sigma_{\min}(C')}\right)^2 \cdot \kappa(H)^5$$

using Lemma D.2 and our choice of $\beta$.

Let us recap what we have done so far. We started with a nonlinear system $(f, h, x_0)$ with an $\varepsilon$-approximate linearization $(A', C', z_0)$ such that $Q_\star(\varepsilon) = Q_\star(A', C', \Sigma) = Q_\star(f, h, \varepsilon, \Sigma)$ as per Definition B.8. We constructed an equivalent $\varepsilon$-approximate linearization $(\bar{A}, \bar{C}, \bar{z}_0)$ with

$$\bar{Q}_\star = Q_\star(\bar{A}, \bar{C}, \Sigma) \lesssim \left(1 + \frac{\sigma_{\max}(C')}{\sigma_{\min}(C')}\right)^2 \cdot Q_\star(\varepsilon)^5$$

and such that $\frac{\sigma_{\max}(\bar{C})}{\sigma_{\min}(\bar{C})} = 1$, $\|\bar{C}\|_F = \sqrt{d_{\mathcal{Y}}}$, and $\|\bar{z}_0\| \leq \|z_0\| + R$. Thus, we may treat $(y_t)_t$ as the outputs of a disturbed version of the LDS $(\bar{A}, \bar{C}, \bar{z}_0)$ with disturbance sizes going to 0 as $\varepsilon \downarrow 0$. More precisely, for any finite $\varepsilon$ we may at each step inject the error $y'_t - y_t$ as process noise into the second coordinate of the hidden states of $(\bar{A}, \bar{C}, \bar{z}_0)$ so that the disturbed rollout produces the true observations $y_t$ and $\|w_t\| = \|y'_t - y_t\|$. In particular, we know $W \leq T^2 \varepsilon$ and applying Lemma B.5 yields $\sum_{t=1}^T \|w_t\|^2 = \sum_{t=1}^T \|y'_t - y_t\|^2 \leq RT^2 \varepsilon$. We apply Theorem A.3 to find that if we denote $\bar{Q} = d_{\mathcal{Y}} \bar{Q}_\star^2$ and $R_0 = R + \frac{\sqrt{d_{\mathcal{Y}}} \bar{Q}_\star T^2 \varepsilon}{\rho}$ then

$$\sum_{t=1}^T \|\hat{y}_t - y_t\|^2 \leq O\left(R_0^2 d_{\mathcal{Y}}^2 \cdot \frac{\log_+(R_0 \bar{Q} T / \gamma)^3}{\gamma} + R_0^2 \cdot \frac{\log_+(\sqrt{d_{\mathcal{Y}}} \bar{Q}_\star R / R_0)}{\rho} + \frac{d_{\mathcal{Y}} \bar{Q}_\star^2}{\rho^2} RT^2 \varepsilon\right).$$

The quantities $R_0$ and $\bar{Q}$ depend on $\varepsilon$, but the predictions $\hat{y}_t$ do not. So, if we assume that $\limsup_{\varepsilon \downarrow 0} Q_\star(\varepsilon) = Q_\star < \infty$ then we may take $\varepsilon \downarrow 0$, in which case $R_0 \to R$ and the third term above vanishes. Noting that $\bar{Q}_\star \lesssim Q_\star^5$ and $\bar{Q} \lesssim Q_\star^{10}$ in the case $d_{\mathcal{Y}} = 1$, the claimed result follows. $\qquad\square$

## C. Proof of Lemma B.4

In this section we provide a proof of Lemma B.4, which follows the proof techniques from (Hazan et al., 2017).

*Proof of Lemma B.4.* We measure regret against $\text{LDS}(\rho, \gamma, \kappa, R_B, R_C, R_{x_0})$, which we recall to be the class of LDS predictors with spectrum $\sigma(A) \subset [0, 1 - \rho] \cup \{1\} \cup \mathbb{D}_{1-\gamma}$, spectral condition number $\kappa_{\text{diag}}(A) \leq \kappa$, and the bounds $\|B\|_F \leq R_B$, $\|C\|_F \leq R_C$, and $\|x_0\| \leq R_{x_0}$. For notation, we let $\Theta = (A, B, C, x_0)$ denote the LDS predictor and $y_t^\Theta$ its predictions, and $A = HDH^{-1}$ with $D = \text{diag}(\alpha_1, \ldots, \alpha_{d_h})$ the eigenvalues of $A$.

We first prove an approximation result, i.e. that there exist parameters $J, M, P, N$ that match the predictions of any given LDS. As in Section 2.2 of (Hazan et al., 2017), our predictor $\hat{y}_{t+1}$ will add a prediction of the derivative of the impulse response $y_{t+1}^\Theta - y_t^\Theta$ to $y_t$. The derivative is

$$y_t^\Theta - y_{t-1}^\Theta = CBu_{t-1} + \sum_{i=1}^t C(A^i - A^{i-1})Bu_{t-i-1} + C(A^t - A^{t-1})x_0$$

$$= CBu_{t-1} + \sum_{i=1}^t CH(D^i - D^{i-1})H^{-1}Bu_{t-i-1} + C(A^t - A^{t-1})x_0.$$

We will order the eigenvalues so that there exists $N \in [d_h]$ such that $\alpha_j \in [0, 1]$ for all $j \leq N$ and $|\alpha_j| \leq 1 - \gamma$ for all

$j > N$. Defining $\mu_\alpha = (\alpha - 1)[1, \alpha, \ldots, \alpha^{T-2}] \in \mathbb{R}^{T-1}$, we have

$$y_t^\Theta - y_{t-1}^\Theta = CBu_{t-1} + \sum_{i=1}^{t} \sum_{j=1}^{d_h} (\alpha_j^i - \alpha_j^{i-1}) CHe_j e_j^\top H^{-1} Bu_{t-i-1} + C(A^t - A^{t-1})x_0$$

$$= CBu_{t-1} + \sum_{j=1}^{d_h} CHe_j e_j^\top H^{-1} B \left( \sum_{i=1}^{t} (\alpha_j - 1)\alpha_j^{i-1} u_{t-i-1} \right) + C(A^t - A^{t-1})x_0$$

$$= CBu_{t-1} + \sum_{j=1}^{d_h} CHe_j e_j^\top H^{-1} B \mu_{\alpha_j}^\top u_{t-2:t-T} + C(A^t - A^{t-1})x_0,$$

where $u_{t-2:t-T} \in \mathbb{R}^{(T-1) \times d_{\text{in}}}$ and so $\mu_{\alpha_j}^\top u_{t-2:t-T} \in \mathbb{R}^{d_{\text{in}}}$. We split the sum over hidden dimension into two terms: $j \leq N$ will be handled via spectral filtering and $j > N$ via regression. Letting $\{\sigma_i, \phi_i\}_{i=1}^{T-1}$ be the eigenvalues and eigenvectors of the Hankel matrix from Algorithm 1 (see Section 3 of (Hazan et al., 2017)), we have

$$\sum_{j=1}^{N} CHe_j e_j^\top H^{-1} \mu_{\alpha_j}^\top u_{t-2:t-T} = \sum_{j=1}^{N} CHe_j e_j^\top H^{-1} B \mu_{\alpha_j}^\top \left( \sum_{i=1}^{T-1} \phi_i \phi_i^\top \right) u_{t-2:t-T}$$

$$= \sum_{i=1}^{T-1} \sum_{j=1}^{N} CHe_j e_j^\top H^{-1} B (\mu_{\alpha_j}^\top \phi_i)(\phi_i^\top u_{t-2:t-T})$$

Define the optimal parameters $J_1 = CB$, $J_i = CH \left( \sum_{j=N+1}^{d_h} (\alpha_j^i - \alpha_j^{i-1}) e_j e_j^\top H^{-1} B \right)$ for $i > 1$, $M_i = \sigma_i^{-1/4} CH \left( \sum_{j=1}^{N} \mu_{\alpha_j}^\top \phi_i e_j e_j^\top \right) H^{-1} B$ for $i \in [T-1]$, and $P_1 = 1$, $P_i = 0$ for $i > 1$, and $N \equiv 0$. Then, we have the approximation error[15]

$$y_t^\Theta - \hat{y}_t(J, M, P, N) = \sum_{i=m+1}^{t} J_i u_{t-i} + \sum_{i=h+1}^{T-1} \sigma_i^{1/4} M_i \langle \phi_i, u_{t-2:t-T} \rangle + C(A^t - A^{t-1})x_0.$$

We bound

$$\|J_i\| \leq \left( \sup_{\alpha \in \mathbb{D}_{1-\gamma}} |\alpha^i - \alpha^{i-1}| \right) \cdot \sum_{j=N+1}^{d_h} \|CHe_j\| \cdot \|e_j^\top H^{-1} B\|$$

$$\leq 2(1-\gamma)^{i-1} \cdot \|CH\|_F \cdot \|H^{-1}B\|_F$$

$$\leq 2(1-\gamma)^{i-1} R_B R_C \kappa,$$

where in the second line we applied Cauchy-Schwartz and in the third we used $\|S_1 S_2\|_F \leq \min(\|S_1\| \|S_2\|_F, \|S_1\|_F \|S_2\|)$. Similarly,

$$\|M_i\| \leq \sigma_i^{-1/4} \cdot \left( \sup_{\alpha \in [0,1]} |\mu_\alpha^\top \phi_i| \right) \cdot \sum_{j=1}^{N} \|CHe_j\| \cdot \|e_j^\top H^{-1} B\|$$

$$\leq 6^{1/4} R_B R_C \kappa,$$

where the second line uses Lemma E.4 of (Hazan et al., 2017). Since $\|u_{t-i}\| \leq R$ and $|\langle \phi_i, u_{t-2} : t - T \rangle| \leq R\sqrt{T}$ by Cauchy-Schwartz, it remains to bound $\|C(A^t - A^{t-1})x_0\|$. If $A$ has all eigenvalues 1 then this term is zero, and so suppose that the largest eigenvalue of $A$ not equal to 1 has magnitude $1 - \rho$. Then,

$$\|C(A^t - A^{t-1})x_0\| \leq \sum_{j=1}^{d_h} |\alpha_j^t - \alpha_j^{t-1}| \cdot \|CHe_j\| \cdot \|e_j^\top H^{-1} x_0\|.$$

---

[15]Technically, we are considering the class of LDS predictors which add the derivative of the impulse response to the ground truth $y_{t-1}$. Otherwise, we would suffer an additional $y_{t-1} - y_{t-1}^\Theta$ approximation error.

For $j$ such that $\alpha_j = 1$ the summand is 0, and for all other $j$ we know $|\alpha_j^t - \alpha_j^{t-1}| \leq 2 \cdot (1-\rho)^{t-1}$. Furthermore, again by Cauchy-Schwartz we have $\sum_{j=1}^{d_h} \|CHe_j\| \cdot \|e_j^\top H^{-1}x_0\| \leq \|CH\|_F \cdot \|H^{-1}x_0\| \leq R_C \kappa R_{x_0}$. Combining these ingredients,

$$\left\|y_t^\Theta - \hat{y}_t(J, M, P, N)\right\| \leq 2R_B R_C \kappa R \cdot \sum_{i=m+1}^{T} (1-\gamma)^{i-1} + 6^{1/4} R_B R_C \kappa R \sqrt{T} \cdot \sum_{i=h+1}^{T-1} \sigma_i^{1/4} + 2R_C \kappa R_{x_0} \cdot (1-\rho)^{t-1}$$

We note that $(1-\gamma)^n \leq e^{-\gamma n}$ and that $\sigma_i^{1/4} \leq K e^{-i/(2\log T)}$ for an absolute constant $K < 10^6$ by Lemma E.2 of (Hazan et al., 2017). Since $\sum_{i>n} e^{-ai} \leq \frac{e^{-an}}{a}$, we get the approximation result

$$\left\|y_t^\Theta - \hat{y}_t(J, M, P, N)\right\| \lesssim R_B R_C \kappa R \cdot \frac{e^{-\gamma(m-1)}}{\gamma} + R_B R_C \kappa R \log(T)\sqrt{T} \cdot e^{-\frac{h}{2\log T}} + R_C \kappa R_{x_0} \cdot e^{-\rho(t-1)}$$

However, we also know that $\left\|y_t^\Theta - \hat{y}_t(J, M, P, N)\right\| \leq 2R$, and so if we let $\tau = \max\left(\frac{\log(R_C \kappa R_{x_0}/2R)}{\rho}, 1\right)$ then $R_C \kappa R_{x_0} e^{-\rho\tau} \leq 2R$, and therefore

$$\left\|y_t^\Theta - \hat{y}_t(J, M, P, N)\right\| \lesssim R_B R_C \kappa R \cdot \frac{e^{-\gamma(m-1)}}{\gamma} + R_B R_C \kappa R \log(T)\sqrt{T} \cdot e^{-\frac{h}{2\log T}} + R \cdot \begin{cases} 1 & t \leq \tau \\ e^{-\rho(t-\tau)} & t > \tau \end{cases}$$

Summing over all $t$ and using $\sum_{t>\tau} e^{-\rho(t-\tau)} \leq \frac{1}{\rho}$ gives

$$\sum_{t=1}^{T} \left\|y_t^\Theta - \hat{y}_t(J, M, P, N)\right\| \lesssim R_B R_C \kappa R T \cdot \frac{e^{-\gamma(m-1)}}{\gamma} + R_B R_C \kappa R \log(T) T^{3/2} \cdot e^{-\frac{h}{2\log T}}$$

$$+ \frac{R \log_+(2R_C \kappa R_{x_0}/R)}{\rho} \ .$$

Taking $m = \Theta\left(\frac{1}{\gamma}\log\left(\frac{R_B R_C \kappa T}{\gamma}\right)\right)$ and $h = \Theta\left(\log(T)\log(R_B R_C \kappa T^{3/2}\log(T))\right)$, we get for any LDS parameterized by $\Theta$ there exist parameters $J_\Theta, M_\Theta, P_\Theta, N_\Theta$ for which

$$\sum_{t=1}^{T} \left\|y_t^\Theta - \hat{y}_t(J, M, P, N)\right\| \leq O\left(\frac{R \log_+(R_C \kappa R_{x_0}/R)}{\rho}\right) \ .$$

Applying boundedness of the observations and predictions, we have the squared loss approximation result

$$\sum_{t=1}^{T} \left\|y_t^\Theta - \hat{y}_t(J, M, P, N)\right\|^2 \leq O\left(\frac{R^2 \log_+(R_C \kappa R_{x_0}/R)}{\rho}\right) \ .$$

We now run the vector-valued Vovk–Azoury–Warmuth forecaster on the derivative labels

$$z_t := y_t - y_{t-1} \in \mathbb{R}^{d_y}.$$

Define the feature vector

$$a_t := \begin{bmatrix} u_{t-1} \\ \vdots \\ u_{t-m} \\ \sigma_1^{1/4}\langle\phi_1, u_{t-2:t-T}\rangle \\ \vdots \\ \sigma_h^{1/4}\langle\phi_h, u_{t-2:t-T}\rangle \end{bmatrix} \in \mathbb{R}^p, \qquad p = d_u(m+h),$$

with zero padding for unavailable past inputs. The VAW forecaster predicts

$$\hat{z}_t = \widehat{W}_t a_t, \qquad \hat{y}_t = y_{t-1} + \hat{z}_t.$$

For the comparator associated with $\Theta$, define

$$W_\Theta := \begin{bmatrix} J_{\Theta,1} & \cdots & J_{\Theta,m} & M_{\Theta,1} & \cdots & M_{\Theta,h} \end{bmatrix} \in \mathbb{R}^{d_y \times p}.$$

Then

$$\widehat{y}_t(J_\Theta, M_\Theta, P_\Theta, N_\Theta) = y_{t-1} + W_\Theta a_t.$$

Our earlier norm bounds ensure that

$$
\begin{aligned}
\|W_\Theta\|_F^2 &= \sum_{i=1}^{m} \|J_{\Theta,i}\|_F^2 + \sum_{i=1}^{h} \|M_{\Theta,i}\|_F^2 \\
&\lesssim R_B^2 R_C^2 \kappa^2 \left( 1 + \sum_{i=1}^{m} (1-\gamma)^{2i} + h \right) \\
&\lesssim R_B^2 R_C^2 \kappa^2 \left( h + \frac{1}{\gamma} \right).
\end{aligned}
$$

Moreover, using the same filtered-feature bound as above,

$$
\begin{aligned}
\|a_t\|^2 &= \sum_{i=1}^{m} \|u_{t-i}\|^2 + \sum_{i=1}^{h} \sigma_i^{1/2} \|\langle \phi_i, u_{t-2:t-T} \rangle\|^2 \\
&\lesssim R^2 (m + h \log^2 T).
\end{aligned}
$$

By the vector-valued Vovk–Azoury–Warmuth regret guarantee of Lemma C.1,

$$\sum_{t=1}^{T} \|\widehat{z}_t - z_t\|^2 - \sum_{t=1}^{T} \|W_\Theta a_t - z_t\|^2 \le \lambda \|W_\Theta\|_F^2 + 4 d_y R^2 \log \det \left( I_p + \frac{1}{\lambda} \sum_{t=1}^{T} a_t a_t^\top \right).$$

Using $\log \det(I + X) \le p \log(1 + \operatorname{tr}(X)/p)$ and taking $\lambda = \Theta\left( \frac{1}{R_B^2 R_C^2 \kappa^2 (h + \frac{1}{\gamma})} \right)$,

$$
\begin{aligned}
&\sum_{t=1}^{T} \|\widehat{y}_t - y_t\|^2 - \sum_{t=1}^{T} \|\widehat{y}_t(J_\Theta, M_\Theta, P_\Theta, N_\Theta) - y_t\|^2 \\
&\lesssim d_y R^2 p \log \left( 1 + \frac{T R^2 R_B^2 R_C^2 \kappa^2 \left( h + \frac{1}{\gamma} \right) (m + h \log^2 T)}{p} \right) \\
&= d_y d_u R^2 (m + h) \log \left( 1 + \frac{T R^2 R_B^2 R_C^2 \kappa^2 \left( h + \frac{1}{\gamma} \right) (m + h \log^2 T)}{d_u (m + h)} \right).
\end{aligned}
$$

It remains to compare the finite-filter derivative comparator to the LDS derivative comparator. Let

$$\Delta_t^\Theta := y_t^\Theta - y_{t-1}^\Theta, \qquad \widetilde{\Delta}_t^\Theta := W_\Theta a_t, \qquad \varepsilon_t := \|\widetilde{\Delta}_t^\Theta - \Delta_t^\Theta\|.$$

The approximation argument above gives

$$\sum_{t=1}^{T} \varepsilon_t^2 \lesssim \frac{R^2 \log_+(R_C \kappa R_{x_0}/R)}{\rho}.$$

Since $\|y_t^\Theta\|, \|y_t\| \le R$, we have

$$\|\Delta_t^\Theta - (y_t - y_{t-1})\| \le 4R.$$

Therefore

$$
\begin{aligned}
&\|\widetilde{\Delta}_t^\Theta - (y_t - y_{t-1})\|^2 - \|\Delta_t^\Theta - (y_t - y_{t-1})\|^2 \\
&\le 8R\varepsilon_t + \varepsilon_t^2.
\end{aligned}
$$

Summing and using the approximation bounds gives

$$\sum_{t=1}^{T} \|\widetilde{\Delta}_t^{\Theta} - (y_t - y_{t-1})\|^2 - \sum_{t=1}^{T} \|\Delta_t^{\Theta} - (y_t - y_{t-1})\|^2 \lesssim \frac{R^2 \log_+(R_C \kappa R_{x_0}/R)}{\rho}.$$

$\square$

This proof has three main differences from that of Theorem 1 of (Hazan et al., 2017). Firstly, we prove things for an algorithm which combines regression terms in order to learn systems with spectrum in $[0, 1] \cup \mathbb{D}_{1-\gamma}$; spectral filtering handles the $[0, 1]$ part and regression with $\widetilde{O}\left(\frac{1}{\gamma}\right)$ parameters handles the $\mathbb{D}_{1-\gamma}$ part. Secondly, we use the VAW forecaster to ensure logarithmic dependence on the optimization diameter.

The third (and more theoretically interesting) difference is that rather than handling the vanishing initial state using the envelope $\sup_{\alpha \in [0,1]} |\alpha^t - \alpha^{t-1}| \leq \frac{1}{t}$ to get linear decay, we make use of a spectral gap condition to get exponential decay of the initial state effect. If we don't want to assume a spectral gap (i.e. $\rho = 0$), then we can instead use that $\sup_{\alpha \in [0,1]} |\alpha^t - \alpha^{t-1}| \leq \frac{1}{t}$ and $\sup_{\alpha \in \mathbb{D}_{1-\gamma}} |\alpha^t - \alpha^{t-1}| \leq 2(1-\gamma)^{t-1}$ to get

$$\sum_{t=1}^{T} \left\| C(A^t - A^{t-1})x_0 \right\| \leq \|C\|\kappa \cdot \left(\frac{1}{\gamma} + \log T\right),$$

which plugs into the approximation part of the above proof.

### C.1. VAW Forecaster

We use the Vovk-Azoury-Warmuth forecaster (Azoury & Warmuth, 2001) for online least squares, and for completeness we prove the regret bound we used in the proof of Lemma B.4.

---
**Algorithm 2** Vector-Valued Vovk-Azoury-Warmuth Forecaster
---
1: **Input:** Regularization $\lambda > 0$
2: Initialize $A_0 = \lambda I_p, \qquad V_0 = 0_{d \times p}$.
3: **for** $t = 1, \ldots, T$ **do**
4:     Observe feature vector $\mathbf{a}_t \in \mathbb{R}^p$
5:     Update covariance: $A_t = A_{t-1} + \mathbf{a}_t \mathbf{a}_t^\top$
6:     Predict vector label: $\hat{\mathbf{z}}_t = V_{t-1} A_t^{-1} \mathbf{a}_t \in \mathbb{R}^d$
7:     Observe true vector label $\mathbf{z}_t \in \mathbb{R}^d$
8:     Update sufficient statistics: $V_t = V_{t-1} + \mathbf{z}_t \mathbf{a}_t^\top$
9: **end for**
---

**Lemma C.1** (Vector-valued VAW regret). *Assume that $\|\mathbf{z}_t\|_\infty \leq Y$ for all $t \in [T]$. Then for any matrix $M \in \mathbb{R}^{d \times p}$,*

$$\sum_{t=1}^{T} \|\hat{\mathbf{z}}_t - \mathbf{z}_t\|_2^2 - \sum_{t=1}^{T} \|M\mathbf{a}_t - \mathbf{z}_t\|_2^2 \leq \lambda\|M\|_F^2 + dY^2 \log\det\left(I_p + \frac{1}{\lambda}\sum_{t=1}^{T}\mathbf{a}_t\mathbf{a}_t^\top\right).$$

*In particular, setting $\lambda = 1/\|M\|_F^2$ and if $\|\mathbf{a}_t\| \leq R$ for all $t \in [T]$ then*

$$\sum_{t=1}^{T} \|\hat{\mathbf{z}}_t - \mathbf{z}_t\|_2^2 - \sum_{t=1}^{T} \|M\mathbf{a}_t - \mathbf{z}_t\|_2^2 \leq \lambda \|M\|_F^2 + dpY^2 \log\left(1 + \frac{TR^2}{\lambda}\right) \leq C_0 dpY^2 \log(TR^2 \|M\|_F^2)$$

*for some universal constant $C_0$.*

*Proof.* Let

$$\mathbf{M} = \begin{bmatrix} \mathbf{m}_1^\top \\ \vdots \\ \mathbf{m}_d^\top \end{bmatrix}, \qquad \mathbf{m}_j \in \mathbb{R}^p, \qquad \mathbf{z}_t = \begin{bmatrix} z_t^{(1)} \\ \vdots \\ z_t^{(d)} \end{bmatrix}, \qquad \hat{\mathbf{z}}_t = \begin{bmatrix} \hat{z}_t^{(1)} \\ \vdots \\ \hat{z}_t^{(d)} \end{bmatrix}.$$

For each coordinate $j \in [d]$, define
$$\mathbf{v}_t^{(j)} := \sum_{s=1}^{t} z_s^{(j)} \mathbf{a}_s \in \mathbb{R}^p.$$

Then by construction of Algorithm 2,
$$\hat{z}_t^{(j)} = \mathbf{a}_t^\top A_t^{-1} \mathbf{v}_{t-1}^{(j)}.$$

We now prove the scalar regret bound for each coordinate directly, and then sum over coordinates. Fix $j \in [d]$. Define the scalar regularized least-squares objectives

$$F_t^{(j)}(\mathbf{x}) := \lambda \|\mathbf{x}\|_2^2 + \sum_{s=1}^{t} (\mathbf{x}^\top \mathbf{a}_s - z_s^{(j)})^2, \qquad \mathbf{x} \in \mathbb{R}^p.$$

Observe by definition of $A_t$ that

$$F_t^{(j)}(\mathbf{x}) = \mathbf{x}^\top A_t \mathbf{x} - 2(\mathbf{v}_t^{(j)})^\top \mathbf{x} + \sum_{s=1}^{t} (z_s^{(j)})^2,$$

and its unique minimizer is
$$\mathbf{x}_{t+1}^{(j)} = A_t^{-1} \mathbf{v}_t^{(j)}.$$

In particular,
$$\mathbf{x}_t^{(j)} = A_{t-1}^{-1} \mathbf{v}_{t-1}^{(j)}.$$

Define
$$r_t^{(j)} := (\mathbf{x}_t^{(j)})^\top \mathbf{a}_t = \mathbf{a}_t^\top A_{t-1}^{-1} \mathbf{v}_{t-1}^{(j)}, \qquad q_t := \mathbf{a}_t^\top A_{t-1}^{-1} \mathbf{a}_t.$$

By the Sherman–Morrison formula,
$$A_t^{-1} \mathbf{a}_t = \frac{A_{t-1}^{-1} \mathbf{a}_t}{1 + q_t},$$

and therefore
$$\hat{z}_t^{(j)} = \mathbf{a}_t^\top A_t^{-1} \mathbf{v}_{t-1}^{(j)} = \frac{r_t^{(j)}}{1 + q_t}.$$

Thus VAW shrinks the ordinary ridge prediction $r_t^{(j)}$ by the factor $(1 + q_t)^{-1}$.

We next compute the one-step increase in the minimum value of the regularized least-squares objective. Since

$$\min_{\mathbf{x}} F_t^{(j)}(\mathbf{x}) = \sum_{s=1}^{t} (z_s^{(j)})^2 - (\mathbf{v}_t^{(j)})^\top A_t^{-1} \mathbf{v}_t^{(j)},$$

we have
$$F_t^{(j)}(\mathbf{x}_{t+1}^{(j)}) - F_{t-1}^{(j)}(\mathbf{x}_t^{(j)}) = (z_t^{(j)})^2 - (\mathbf{v}_t^{(j)})^\top A_t^{-1} \mathbf{v}_t^{(j)} + (\mathbf{v}_{t-1}^{(j)})^\top A_{t-1}^{-1} \mathbf{v}_{t-1}^{(j)}.$$

Now write $\mathbf{v}_t^{(j)} = \mathbf{v}_{t-1}^{(j)} + z_t^{(j)} \mathbf{a}_t$, and use

$$A_t^{-1} = A_{t-1}^{-1} - \frac{A_{t-1}^{-1} \mathbf{a}_t \mathbf{a}_t^\top A_{t-1}^{-1}}{1 + q_t}.$$

A direct expansion gives

$$(\mathbf{v}_t^{(j)})^\top A_t^{-1} \mathbf{v}_t^{(j)} = (\mathbf{v}_{t-1}^{(j)})^\top A_{t-1}^{-1} \mathbf{v}_{t-1}^{(j)} + (z_t^{(j)})^2 - \frac{(z_t^{(j)} - r_t^{(j)})^2}{1 + q_t}.$$

Hence
$$F_t^{(j)}(\mathbf{x}_{t+1}^{(j)}) - F_{t-1}^{(j)}(\mathbf{x}_t^{(j)}) = \frac{(z_t^{(j)} - r_t^{(j)})^2}{1 + q_t}. \tag{1}$$

We now compare the VAW loss to this increment. Since

$$\hat{z}_t^{(j)} - z_t^{(j)} = \frac{r_t^{(j)}}{1 + q_t} - z_t^{(j)} = \frac{(r_t^{(j)} - z_t^{(j)}) - z_t^{(j)} q_t}{1 + q_t},$$

we obtain

$$(\hat{z}_t^{(j)} - z_t^{(j)})^2 = \frac{((r_t^{(j)} - z_t^{(j)}) - z_t^{(j)} q_t)^2}{(1 + q_t)^2}.$$

Using the elementary inequality

$$(u - vq)^2 \leq (1 + q)u^2 + (1 + q)v^2 q \qquad \text{for all } u, v \in \mathbb{R}, \ q \geq 0,$$

with $u = r_t^{(j)} - z_t^{(j)}$ and $v = z_t^{(j)}$, we get

$$(\hat{z}_t^{(j)} - z_t^{(j)})^2 \leq \frac{(r_t^{(j)} - z_t^{(j)})^2}{1 + q_t} + (z_t^{(j)})^2 \frac{q_t}{1 + q_t}.$$

Since

$$\mathbf{a}_t^\top A_t^{-1} \mathbf{a}_t = \frac{q_t}{1 + q_t},$$

and by (1),

$$(\hat{z}_t^{(j)} - z_t^{(j)})^2 \leq F_t^{(j)}(\mathbf{x}_{t+1}^{(j)}) - F_{t-1}^{(j)}(\mathbf{x}_t^{(j)}) + (z_t^{(j)})^2 \, \mathbf{a}_t^\top A_t^{-1} \mathbf{a}_t.$$

Summing over $t = 1, \ldots, T$ telescopes the objective terms:

$$\sum_{t=1}^{T}(\hat{z}_t^{(j)} - z_t^{(j)})^2 \leq F_T^{(j)}(\mathbf{x}_{T+1}^{(j)}) - F_0^{(j)}(\mathbf{x}_1^{(j)}) + \sum_{t=1}^{T}(z_t^{(j)})^2 \, \mathbf{a}_t^\top A_t^{-1} \mathbf{a}_t.$$

Because $F_0^{(j)}(\mathbf{x}) = \lambda\|\mathbf{x}\|_2^2$ is minimized at $\mathbf{x}_1^{(j)} = 0$, we have $F_0^{(j)}(\mathbf{x}_1^{(j)}) = 0$. By optimality of $\mathbf{x}_{T+1}^{(j)}$,

$$F_T^{(j)}(\mathbf{x}_{T+1}^{(j)}) \leq F_T^{(j)}(\mathbf{m}_j) = \lambda\|\mathbf{m}_j\|_2^2 + \sum_{t=1}^{T}(\mathbf{m}_j^\top \mathbf{a}_t - z_t^{(j)})^2.$$

Therefore

$$\sum_{t=1}^{T}(\hat{z}_t^{(j)} - z_t^{(j)})^2 - \sum_{t=1}^{T}(\mathbf{m}_j^\top \mathbf{a}_t - z_t^{(j)})^2 \leq \lambda\|\mathbf{m}_j\|_2^2 + \sum_{t=1}^{T}(z_t^{(j)})^2 \, \mathbf{a}_t^\top A_t^{-1} \mathbf{a}_t.$$

Since $|z_t^{(j)}| \leq \|\mathbf{z}_t\|_\infty \leq Y$, this yields

$$\sum_{t=1}^{T}(\hat{z}_t^{(j)} - z_t^{(j)})^2 - \sum_{t=1}^{T}(\mathbf{m}_j^\top \mathbf{a}_t - z_t^{(j)})^2 \leq \lambda\|\mathbf{m}_j\|_2^2 + Y^2 \sum_{t=1}^{T} \mathbf{a}_t^\top A_t^{-1} \mathbf{a}_t. \qquad (2)$$

It remains to bound the shared covariance term. Let

$$u_t := \mathbf{a}_t^\top A_t^{-1} \mathbf{a}_t.$$

Since $A_t = A_{t-1} + \mathbf{a}_t \mathbf{a}_t^\top$, the matrix determinant lemma gives

$$\det(A_t) = \det(A_{t-1})\big(1 + \mathbf{a}_t^\top A_{t-1}^{-1} \mathbf{a}_t\big).$$

But

$$u_t = \frac{\mathbf{a}_t^\top A_{t-1}^{-1} \mathbf{a}_t}{1 + \mathbf{a}_t^\top A_{t-1}^{-1} \mathbf{a}_t},$$

so

$$1 - u_t = \frac{1}{1 + \mathbf{a}_t^\top A_{t-1}^{-1} \mathbf{a}_t}, \qquad -\log(1 - u_t) = \log \frac{\det(A_t)}{\det(A_{t-1})}.$$

Since $-\log(1-u) \geq u$ for all $u \in [0,1)$, we obtain

$$u_t \leq \log \frac{\det(A_t)}{\det(A_{t-1})}.$$

Summing over $t$ gives

$$\sum_{t=1}^{T} \mathbf{a}_t^\top A_t^{-1} \mathbf{a}_t \leq \log \frac{\det(A_T)}{\det(A_0)} = \log \det \left( I_p + \frac{1}{\lambda} \sum_{t=1}^{T} \mathbf{a}_t \mathbf{a}_t^\top \right). \tag{3}$$

Finally, sum (2) over $j = 1, \ldots, d$. Using

$$\|\hat{\mathbf{z}}_t - \mathbf{z}_t\|_2^2 = \sum_{j=1}^{d} (\hat{z}_t^{(j)} - z_t^{(j)})^2, \qquad \|\mathbf{M}\mathbf{a}_t - \mathbf{z}_t\|_2^2 = \sum_{j=1}^{d} (\mathbf{m}_j^\top \mathbf{a}_t - z_t^{(j)})^2,$$

and

$$\|\mathbf{M}\|_F^2 = \sum_{j=1}^{d} \|\mathbf{m}_j\|_2^2,$$

we conclude that

$$\sum_{t=1}^{T} \|\hat{\mathbf{z}}_t - \mathbf{z}_t\|_2^2 - \sum_{t=1}^{T} \|\mathbf{M}\mathbf{a}_t - \mathbf{z}_t\|_2^2 \leq \lambda \|\mathbf{M}\|_F^2 + dY^2 \sum_{t=1}^{T} \mathbf{a}_t^\top A_t^{-1} \mathbf{a}_t.$$

Applying (3) completes the proof:

$$\sum_{t=1}^{T} \|\hat{\mathbf{z}}_t - \mathbf{z}_t\|_2^2 - \sum_{t=1}^{T} \|\mathbf{M}\mathbf{a}_t - \mathbf{z}_t\|_2^2 \leq \lambda \|\mathbf{M}\|_F^2 + dY^2 \log \det \left( I_p + \frac{1}{\lambda} \sum_{t=1}^{T} \mathbf{a}_t \mathbf{a}_t^\top \right).$$

$\square$

## D. Luenberger Program

In this section, we investigate the Luenberger program of Definition B.1, restated here for convenience: for given $A \in \mathbb{R}^{d_h \times d_h}$, $C \in \mathbb{R}^{d_{\text{obs}} \times d_h}$, and $\Sigma \subset \mathbb{D}_1$, we consider the optimization

$$\min_{L \in \mathbb{R}^{d_h \times d_{\text{obs}}}} \kappa_{\text{diag}}(A - LC) \quad \text{s.t.} \quad A - LC \text{ has all eigenvalues in } \Sigma, \tag{D.1}$$

and we denote the minimal value by $Q_\star = Q(A, C, \Sigma)$ and a minimizing matrix, when it exists, by $L_\star$. First, we note that observability is sufficient for well-posedness:

**Lemma D.1.** *Suppose that $(A, C)$ is observable. Then, for any monic, degree-$d_h$ polynomial $p$ with distinct roots, there is a unique $L \in \mathbb{R}^{d_h \times d_{\text{obs}}}$ such that $A - LC$ is diagonalizable with characteristic polynomial $p$. In particular, if $(A, C)$ is observable then $Q_\star(A, C, \Sigma) < \infty$ for any $\Sigma$ which contains $d_h$ points that are symmetric about the real line.*

*Proof.* This is a standard result in eigenvalue placement, and for $d_{\text{obs}} = 1$ there is an explicit formula known as Ackermann's formula (see e.g. Corollary 2.6 of (Mehrmann & Xu, 1996)). $\square$

The above shows that observability implies $Q_\star < \infty$. Also, the bijection between $p$ and $L$ informs us that we may instead consider the optimization (D.1) as being parameterized by the desired poles. Next, the reader may ask whether our choice of $\kappa_{\text{diag}}(A - LC)$ is the right quantity to capture the "size/complexity" of the observer system as opposed to e.g. the norm of the gain $L$. The following result convinces us that it is:

**Lemma D.2.** *Suppose that $(A, C)$ is observable and let $Q_\star = Q_\star(A, C, \Sigma)$ with $L_\star$ a minimizing gain. Then,*

$$\|L_\star\| \leq \frac{2Q_\star}{\sigma_{min}(C)}.$$

*Proof.* This is equation (22) of (Kautsky et al., 1985) combined with the fact that $Q_\star \geq 1$. $\square$

Lastly, we provide a crude but general upper bound on $Q_\star$ in terms of the number of system eigenvalues which need to be moved:

**Lemma D.3.** *Let $d_{\mathcal{Y}} = 1$. Suppose that $(A, C)$ is observable and $\Sigma \subset \mathbb{D}_1$ is Ahlfors regular with nonempty interior, and let $Av_i = \lambda_i v_i$ denote the eigenvalues and eigenvectors of $A$. Let $S := \{i : \ \lambda_i \notin \Sigma\}$ be the set of undesirable eigenvalues and define*

$$\delta := \min_{\substack{i,j \in S \\ i \neq j}} |\lambda_i - \lambda_j|$$

*to be the minimal gap between undesirable eigenvalues. Then, with $N = |S|$ we have the bound*

$$Q_\star(A, C, \Sigma) \leq \kappa_{diag}(A) \cdot \delta^{-N} \cdot \left(1 + \frac{N}{\text{vol}(\Sigma)}\right)^N \cdot \frac{\max_{i \in S} |Cv_i|}{\min_{i \in S} |Cv_i|}.$$

*Remark* D.4. So, if the undesirable eigenvalues are (1) well-spaced (pairwise distances $\asymp N^{-1}$) and (2) observable with conditioning $N^{O(N)}$, we have $Q_\star(A, C, \Sigma) \lesssim N^{O(N)}$.

*Proof.* Let $U_{\text{in}}$ be the direct sum of eigenspaces of $A$ whose eigenvalues already lie in $\Sigma$, and let $U_{\text{out}}$ be the complementary $A$-invariant subspace, of dimension $N$, spanned by the remaining eigenspaces. In a basis adapted to the decomposition $\mathbb{R}^{d_h} = U_{\text{in}} \oplus U_{\text{out}}$, we may write

$$A = \begin{bmatrix} A_{\text{in}} & 0 \\ 0 & A_{\text{out}} \end{bmatrix}, \qquad C = \begin{bmatrix} C_{\text{in}} & C_{\text{out}} \end{bmatrix}.$$

For the purpose of proving an upper bound, we may restrict attention to gains of the form

$$L = \begin{bmatrix} 0 \\ L_{\text{out}} \end{bmatrix},$$

for which

$$A - LC = \begin{bmatrix} A_{\text{in}} & 0 \\ -L_{\text{out}}C_{\text{in}} & A_{\text{out}} - L_{\text{out}}C_{\text{out}} \end{bmatrix}.$$

Hence $A - LC$ is block lower triangular, so its spectrum is the union of $\sigma(A_{\text{in}}) \subseteq \Sigma$ and $\sigma(A_{\text{out}} - L_{\text{out}}C_{\text{out}})$. Therefore only the $N \times N$ block $A_{\text{out}}$ corresponding to the eigenvalues of $A$ outside $\Sigma$ needs to be modified. Thus, without loss of generality, we may assume that $A$ is $N \times N$.

Since $d_{\mathcal{Y}} = 1$ and $(A, C)$ is observable, every eigenspace of $A$ is one-dimensional and $Cv_i \neq 0$. Write

$$V = [v_i]_{i \in S}, \qquad D_C = \text{diag}(Cv_i)_{i \in S}.$$

The observability matrix is

$$\mathcal{O} = \begin{bmatrix} C \\ CA \\ \vdots \\ CA^{N-1} \end{bmatrix}.$$

Let $\mathsf{V}(\lambda)$ denote the Vandermonde matrix

$$\mathsf{V}(\lambda) = [\lambda_i^{k-1}]_{k=1,\dots,N;\, i \in S}.$$

Then

$$\mathcal{O}V = \mathsf{V}(\lambda)D_C.$$

Therefore

$$\kappa(\mathcal{O}) \leq \kappa(V)\kappa(\mathsf{V}(\lambda))\frac{\max_{i \in S} |Cv_i|}{\min_{i \in S} |Cv_i|}.$$

Because $V$ is a column submatrix of the eigenvector matrix of the original $A$,

$$\kappa(V) \leq \kappa_{\text{diag}}(A).$$

We use the following elementary Vandermonde bound. If $x_1, \ldots, x_N \in \mathbb{D}_1$ are distinct and

$$\Delta := \min_{i \neq j} |x_i - x_j|,$$

then

$$\kappa(\mathsf{V}(x)) \leq N^2 2^{N-1} \Delta^{-(N-1)}.$$

Indeed, $\|\mathsf{V}(x)\| \leq \|\mathsf{V}(x)\|_F \leq N$. The $j$-th Lagrange polynomial

$$\ell_j(z) = \prod_{k \neq j} \frac{z - x_k}{x_j - x_k}$$

has denominator at least $\Delta^{N-1}$, and since $|x_k| \leq 1$, every coefficient of the numerator is bounded by $2^{N-1}$. Thus every entry of $\mathsf{V}(x)^{-1}$ is at most $2^{N-1} \Delta^{-(N-1)}$ in absolute value, so

$$\|\mathsf{V}(x)^{-1}\|_2 \leq N 2^{N-1} \Delta^{-(N-1)}.$$

This proves the bound.

Applying this to the source eigenvalues gives

$$\kappa(\mathsf{V}(\lambda)) \leq N^2 2^{N-1} \delta^{-(N-1)}.$$

Hence

$$\kappa(\mathcal{O}) \leq N^2 2^{N-1} \kappa_{\mathrm{diag}}(A) \frac{\max_{i \in S} |C v_i|}{\min_{i \in S} |C v_i|} \delta^{-(N-1)}.$$

Now choose distinct target poles $\mu_1, \ldots, \mu_N \in \Sigma$. By Ahlfors regularity, $\Sigma$ contains $N$ points with separation

$$\delta_\Sigma := \min_{i \neq j} |\mu_i - \mu_j| \geq c_\Sigma \left( \frac{\mathrm{vol}(\Sigma)}{N} \right)^{1/2},$$

where $c_\Sigma > 0$ depends only on the regularity constants. Since $\Sigma \subset \mathbb{D}_1$, this implies the coarser bound

$$\delta_\Sigma^{-1} \leq C_\Sigma \left( 1 + \frac{N}{\mathrm{vol}(\Sigma)} \right)^{1/2}.$$

Therefore, by the same Vandermonde estimate,

$$\kappa(\mathsf{V}(\mu)) \leq N^2 2^{N-1} \left[ C_\Sigma \left( 1 + \frac{N}{\mathrm{vol}(\Sigma)} \right)^{1/2} \right]^{N-1}.$$

Since $(A, C)$ is observable, by Ackermann pole placement there exists $L$ such that $A - LC$ has eigenvalues $\mu_1, \ldots, \mu_N$. In observable canonical coordinates, the closed-loop matrix is a companion matrix with characteristic polynomial $\prod_{j=1}^N (z - \mu_j)$; this companion matrix is diagonalized by a Vandermonde matrix in the target poles. Conjugating back by the observability map gives

$$\kappa_{\mathrm{diag}}(A - LC) \leq c^N \kappa(\mathcal{O}) \kappa(\mathsf{V}(\mu)),$$

for a universal constant $c > 0$. Combining the source and target Vandermonde bounds yields

$$Q_\star(A, C, \Sigma) \leq \kappa_{\mathrm{diag}}(A) \frac{\max_{i \in S} |C v_i|}{\min_{i \in S} |C v_i|} \delta^{-N} \left( 1 + \frac{N}{\mathrm{vol}(\Sigma)} \right)^{O(N)}.$$

$\square$

The above bound is very general, but is exponential in the number of undesirable eigenvalues (in the worst case, exponential in hidden dimension). We now look into a representative example of an asymmetric system to demonstrate the finer properties of this optimization as well as the tightness of Lemma D.3:

*Example* D.5 (Permutation system). Let $A$ be a cyclical $n \times n$ permutation matrix, i.e.

$$
A = \begin{bmatrix}
0 & 1 & 0 & \cdots & 0 \\
0 & 0 & 1 & \cdots & 0 \\
\vdots & \vdots & \ddots & \ddots & \vdots \\
0 & 0 & \cdots & 0 & 1 \\
1 & 0 & \cdots & 0 & 0
\end{bmatrix},
$$

$C = e_1$ be the projection on the first coordinate, and $p$ be a monic degree-$n$ polynomial (with coefficients $p_j$ and roots $\lambda_j$). Then, the gain $L \in \mathbb{R}^{n \times 1}$ and eigenvector matrix $H$ satisfy

$$
\|L\|^2 = \sum_{j=0}^{n} |p_j|^2, \quad \kappa(H) = \kappa\left(\mathrm{diag}\left(\frac{1}{1 - \vec{\lambda}^n}\right) \cdot V(\vec{\lambda})\right)
$$

for $V(\vec{\lambda})$ the Vandermonde matrix with knots $\lambda_j$. In particular, $\|L\|$ can be made to be small but $\kappa(H)$ is always exponential if $\Sigma$ does not include the roots of unity, i.e. if $\Sigma \subset [0, 1 - \rho] \cup \{1\} \cup \mathbb{D}_{1-\gamma}$ then

$$
Q_\star(A, C, \Sigma) = \Omega(2^n).
$$

*Proof.* We will use the explicit formulas for $H$ and $L$ from Corollaries 2.5 and 2.6 of (Mehrmann & Xu, 1996). We know that $c^\top A^k e_j = \langle c, A^k e_j \rangle = \langle c, e_{j+k \mod n} \rangle = \delta_{j+k \mod n = 1}$, and so

$$
CA = e_2, \quad \ldots, \quad CA^{n-1} = e_n
$$

This means that the observability matrix is

$$
\mathcal{O} = \begin{bmatrix} C \\ CA \\ \vdots \\ CA^{n-1} \end{bmatrix} = \begin{bmatrix} e_1 \\ e_2 \\ \vdots \\ e_n \end{bmatrix} = \mathrm{Id}
$$

In particular, since we know that $A$ is unitarily diagonalized as $A = FDF^{-1}$ for $F$ the DFT matrix and $D = \mathrm{diag}(1, \omega, \omega^2, \ldots, \omega^{n-1})$ with $\omega = e^{2\pi i / n}$, we find

$$
L = p(A) \cdot e_n = F \, \mathrm{diag}(1, p(\omega), p(\omega^2), \ldots, p(\omega^{n-1})) F^{-1} \cdot e_n
$$

Writing this in the standard basis, we find that the $j^{th}$ coordinate of $L$ for $1 \leq j \leq n$ is

$$
\sum_{k=0}^{n-1} \frac{\omega^{jk} \cdot p(\omega^k)}{n}
$$

and so

$$
\|L\|^2 = \frac{1}{n^2} \sum_{j=1}^{n} \sum_{k_1=0}^{n-1} \sum_{k_2=0}^{n-1} \omega^{j(k_1+k_2)} \cdot p(\omega^{k_1}) \cdot p(\omega^{k_2})
$$

$$
= \frac{1}{n^2} \sum_{k_1=0}^{n-1} \sum_{k_2=0}^{n-1} p(\omega^{k_1}) \cdot p(\omega^{k_2}) \cdot \sum_{j=1}^{n} \omega^{j(k_1+k_2)}
$$

We note that $\sum_{j=1}^n \omega^{q \cdot j}$ is equal to 0 unless $q$ is a multiple of $n$, in which case it equals $n$. Therefore, we are left with

$$
\begin{aligned}
\|L\|^2 &= \frac{1}{n} \sum_{k_1 + k_2 = n} p(\omega^{k_1}) \cdot p(\omega^{k_2}) \\
&= \frac{1}{n} \sum_{k=0}^{n-1} p(\omega^k) \cdot p(\omega^{n-k}) \\
&= \frac{1}{n} \sum_{k=0}^{n-1} |p(\omega^k)|^2,
\end{aligned}
$$

where we used that $p(\omega^{-k}) = \overline{p(\omega^k)}$ for a polynomial $p$ with real coefficients. Lastly, noting that

$$
p(\omega^k) = \sum_{j=0}^n p_j \omega^{kj}
$$

we see that the sequence $p(\omega^k)$ is none other than the DFT of the sequence $p_j$ of coefficients of $p$. By Parseval's identity, we get the stated result about $L$.

For $H$, we once again write $A = FDF^{-1}$, and so that $j^{th}$ eigenvector $h_j$ of $A - LC$ equals

$$
h_j = (A - \lambda_j \operatorname{Id})^{-1} C = F \operatorname{diag}\left( \frac{1}{1 - \lambda_j}, \frac{1}{\omega - \lambda_j}, \dots, \frac{1}{\omega^{n-1} - \lambda_j} \right) F^{-1} e_1
$$

Noting that $F^{-1} e_1 = \frac{1}{\sqrt{n}}[1, \dots, 1]$, we see that if $V'$ is the Cauchy matrix with entries

$$
V'_{ij} = \frac{1}{\omega^{j-1} - \lambda_i} \quad (1 \leq i, j \leq n)
$$

then $H = \frac{1}{\sqrt{n}} FV$. Therefore, the condition number of $H$ equals that of $V'$. By equation (7) of (Pan, 2013), we find that $V'$ has the same condition number as $(D^n - \operatorname{Id})^{-1} V$, where $V$ is the Vandermonde matrix with knots $\lambda_j$ and $D$ is the diagonal matrix with entries $\lambda_j$. We have arrived at the fact that

$$
Q_\star = \min_{\vec{\lambda} \subset \Sigma} \kappa\left( \operatorname{diag}\left( \frac{1}{1 - \vec{\lambda}^n} \right) \cdot V(\vec{\lambda}) \right).
$$

The only Vandermonde matrices with subexponential condition number are those with knots close to the roots of unity (Pan, 2015), in which case the row rescalings blow up. As such, it can be seen that for any admissible choice of $\lambda_j$'s, the condition number of this row-rescaled Vandermonde matrix will be exponential in the dimension. $\qquad \square$

# E. Lower Bounds

In this section we give lower bounds for learning sequences generated by linear and nonlinear dynamical systems, which complement our upper bounds and show them to be tight in certain respects. This does not mean they cannot be improved: to the contrary, this investigation motivates further study of the leading order constants and parameters of our bounds.

### E.1. LDS with Noise

We have shown that spectral filtering with observation feedback is able to learn asymmetric marginally stable LDS's under adversarial disturbances. Our regret guarantee scales with two quantities: the optimal observer complexity $Q_\star$ and the disturbance sizes $\sum_{t=0}^{T-1} \|w_t\|$. In this section, we investigate the the leading order terms of this result via lower bounds for sequence prediction in terms of these quantities. In particular, we prove that any algorithm which is not able to see $w_t$ when predicting $\hat{y}_{t+1}$ suffers a linear cost in terms of the disturbance sizes and ambient dimension up to constant factors:

**Proposition E.1.** *Let $\mathcal{A}$ be any algorithm that predicts $\hat{y}_{t+1}$ using $u_1, \dots, u_t$, $y_1, \dots, y_t$, and $w_1, \dots, w_{t-1}$. Then, there exists a problem instance $(A, B, C, x_0)$ and sequences $u_1, \dots, u_T$ and $w_1, \dots, w_T$ satisfying Assumption A.1 for which*

$$
\sum_{t=1}^T \|\hat{y}_t^{\mathcal{A}} - y_t\| \geq \Omega\left( d + d \sum_{t=0}^{T-1} \|Cw_t\| \right)
$$

*Proof.* Assume $T \geq d + 1$; otherwise replace $d$ by $\min\{d, T\}$ throughout. We use the convention

$$x_{t+1} = Ax_t + w_t, \qquad y_{t+1} = Cx_{t+1},$$

and the learner predicts $\hat{y}_{t+1}$ before seeing $w_t$.

Let $B = 0$, $u_t \equiv 0$, and $x_0 = 0$. Fix $M \geq 1$. Let $\sigma \in \{\pm 1\}^d$ be chosen uniformly subject to having exactly $d/2$ positive and $d/2$ negative entries, and let $\pi$ be a uniformly random permutation of $[d]$ with $\pi_1 = 1$. Define $C = e_1^\top$ and let $A = A_\pi$ be the cyclic permutation satisfying

$$e_1^\top A^{j-1} = e_{\pi_j}^\top, \qquad j = 1, \ldots, d.$$

Set

$$w_0 = M\sigma, \qquad w_t = 0 \quad \text{for } t \geq 1.$$

Then, for $j = 1, \ldots, d$,

$$y_j = CA^{j-1}w_0 = M\sigma_{\pi_j}.$$

Moreover,

$$\sum_{t=0}^{T-1} \|Cw_t\| = \|Cw_0\| = M.$$

We lower bound the expected loss over the random choice of $(\sigma, \pi)$. At round $j$, even after conditioning on all past observations and on the revealed disturbance vector $w_0$, the next output is the value of a uniformly random unrevealed coordinate. If among the remaining coordinates there are $p_j$ positives and $n_j$ negatives, then for any prediction $a$,

$$\mathbb{E}\left[|a - M\sigma_{\pi_j}| \mid \mathcal{F}_{j-1}, w_0\right] \geq 2M \frac{\min\{p_j, n_j\}}{p_j + n_j}.$$

For $j \leq d/4$, we have $p_j + n_j \geq 3d/4$. Since the initial signs are balanced and the revealed coordinates are a random sample without replacement,

$$\mathbb{E}|p_j - n_j| \leq \sqrt{d}.$$

Hence, for all sufficiently large $d$,

$$\mathbb{E} \frac{\min\{p_j, n_j\}}{p_j + n_j} = \frac{1}{2} - \mathbb{E} \frac{|p_j - n_j|}{2(p_j + n_j)} \geq c$$

for a universal constant $c > 0$. Therefore

$$\mathbb{E} \sum_{j=1}^{d/4} |\hat{y}_j^{\mathcal{A}} - y_j| \geq cMd.$$

By the probabilistic method, there exists a deterministic choice of $(\sigma, \pi)$ such that

$$\sum_{t=1}^{T} |\hat{y}_t^{\mathcal{A}} - y_t| \geq cMd.$$

For this instance,

$$d + d \sum_{t=0}^{T-1} \|Cw_t\| = d + dM \leq 2dM$$

since $M \geq 1$. Thus

$$\sum_{t=1}^{T} |\hat{y}_t^{\mathcal{A}} - y_t| \geq \Omega \left( d + d \sum_{t=0}^{T-1} \|Cw_t\| \right).$$

$\square$

## E.2. Computational Lower Bounds

The preceding lower bound applies to linear dynamical systems with adversarial (or even stochastic) noise. We next give a computational lower bound sketch, based on cryptographic hardness assumptions, for deterministic nonlinear dynamical systems. Note that since the following system operates on a discrete state space it is equal to its Markov chain linearization (equivalently, we may say the dynamics are 1-Lipschitz w.r.t. a discrete metric and apply Lemma B.5 and the rest of our arguments as usual).

**Proposition E.2.** *Let $\{G_d : \{0,1\}^d \to \{0,1\}^{T(d)}\}_{d \geq 1}$ be a PRG secure against nonuniform polynomial-size randomized distinguishers, where $T(d) = \mathrm{poly}(d)$. For $s \in \{0,1\}^d$, write*

$$Y_i(s) := 2G_d(s)_i - 1 \in \{-1, +1\}, \qquad i = 0, \ldots, T(d) - 1.$$

*Consider the deterministic dynamical system*

$$\mathcal{X}_d = \{0,1\}^d \times \{0, \ldots, T(d) - 1\}, \qquad F(s, i) = (s, i + 1 \bmod T(d)),$$

*with observation map*

$$g(s, i) = Y_i(s).$$

*The initial state is $(s, 0)$ for uniformly random $s \sim \mathrm{Unif}(\{0,1\}^d)$.*

*Then for every polynomial-size randomized predictor family $\widehat{f} = \{\widehat{f}_{d,t}\}$, where*

$$\widehat{f}_{d,t} : \{-1, +1\}^{t+1} \to \mathbb{R}$$

*predicts $Y_{t+1}$ from $(Y_0, \ldots, Y_t)$, and for every polynomial $p$, for all sufficiently large $d$ and every $t \in \{0, \ldots, T(d) - 2\}$,*

$$\mathbb{E}_{s,\widehat{f}} \left[ \left| \widehat{f}_{d,t}(Y_0(s), \ldots, Y_t(s)) - Y_{t+1}(s) \right| \right] \geq 1 - \frac{1}{p(d)}.$$

*Equivalently, no efficient predictor can achieve non-negligible advantage over the baseline loss 1 before seeing a full period.*

*Moreover, after observing $(Y_0, \ldots, Y_{T(d)-1})$, there is an $O(2^d \mathrm{poly}(T(d)))$-time algorithm that finds a seed $s' \in \{0,1\}^d$ satisfying $G_d(s') = (G_d(s)_0, \ldots, G_d(s)_{T(d)-1})$, and then predicts all future observations exactly.*

*Proof.* First note that we may clip the predictor output to $[-1, 1]$, since for $y \in \{-1, +1\}$ the projection $\Pi_{[-1,1]}(a)$ satisfies

$$|\Pi_{[-1,1]}(a) - y| \leq |a - y|.$$

Thus assume without loss of generality that $\widehat{f}_{d,t} \in [-1, 1]$. Suppose the claim fails. Then there exist a polynomial $p$, infinitely many $d$, and indices $t_d \in \{0, \ldots, T(d) - 2\}$ such that

$$\mathbb{E}\left[ \left| \widehat{f}_{d,t_d}(Y_0, \ldots, Y_{t_d}) - Y_{t_d+1} \right| \right] \leq 1 - \frac{1}{p(d)}.$$

For $a \in [-1, 1]$ and $y \in \{-1, +1\}$,

$$|a - y| = 1 - ay.$$

Therefore,

$$\mathbb{E}\left[ \widehat{f}_{d,t_d}(Y_0, \ldots, Y_{t_d}) Y_{t_d+1} \right] \geq \frac{1}{p(d)}.$$

We construct a nonuniform randomized distinguisher $D_d$ for $G_d$ by hardwiring $t_d$. Given $z \in \{0,1\}^{T(d)}$, define $Z_i := 2z_i - 1$. Let $a := \widehat{f}_{d,t_d}(Z_0, \ldots, Z_{t_d})$.. The distinguisher outputs 1 with probability

$$\frac{1 + aZ_{t_d+1}}{2}.$$

If $z = G_d(s)$ for uniform $s$, then

$$\Pr[D_d(G_d(s)) = 1] = \frac{1}{2} + \frac{1}{2}\mathbb{E}\left[ \widehat{f}_{d,t_d}(Y_0, \ldots, Y_{t_d}) Y_{t_d+1} \right] \geq \frac{1}{2} + \frac{1}{2p(d)}.$$

If $z$ is uniform in $\{0, 1\}^{T(d)}$, then $Z_{t_d+1}$ is independent of $(Z_0, \ldots, Z_{t_d})$ and has mean zero, so

$$\Pr[D_d(z) = 1] = \frac{1}{2}.$$

Thus $D_d$ distinguishes $G_d(U_d)$ from uniform with advantage at least $1/(2p(d))$, contradicting PRG security. This proves the prediction lower bound.

For the exhaustive-search claim, given the full period $(Y_0, \ldots, Y_{T(d)-1})$, form the bit string

$$b_i = \frac{Y_i + 1}{2}.$$

Enumerate all $s' \in \{0, 1\}^d$ and compute $G_d(s')$ until finding one with $G_d(s') = b$. Such an $s'$ exists because the data was generated by some seed $s$. This takes $O(2^d \text{poly}(T(d)))$ time. Since the dynamics is periodic with period $T(d)$, future observations are

$$Y_\tau = 2G_d(s')_{\tau \bmod T(d)} - 1,$$

so this seed predicts all future observations exactly. $\qquad\square$

## F. Multi-dimensional Observations

In this section, we discuss what changes when allowing higher $d_{\mathcal{Y}}$. To begin, in such settings we can have $(A, C)$ being observable even if $A$ has an eigenvalue with multiplicity $d_{\mathcal{Y}}$, and so eigenvalue placement will allow us to move undesirable eigenvalues with multiplicity. In addition, while Ackermann's formula as stated no longer holds, the bound

$$\|L_\star\| \le \frac{2Q_\star}{\sigma_{\min}(C)}$$

of Lemma D.2 continues to hold. The regret guarantee of Lemma B.4 depends on $\|L_\star\|_F \cdot \|C\|_F$ in a sharp way, and we use that $\|L_\star\|_F = \|L_\star\|$ and $\|C\|_F = \|C\| = \sigma_{\min}(C)$ in the 1-dimensional observation case in order to simplify several expressions. While in most places our bounds will simply introduce a $\sqrt{d_{\mathcal{Y}}}$ factor gotten from converting between operator and Frobenius norms, we will also see a $\frac{\sigma_{\max}(C)}{\sigma_{\min}(C)}$ conditioning factor in the loss bounds of Theorems 4.1 and A.3. Lastly, one would require a multidimensional generalization of Lemma D.3: it still holds that $Q_\star \lesssim \kappa(\mathcal{O}) \cdot (N/\text{vol}(\Sigma))^{O(N)}$ where $\mathcal{O}$ is the multidimensional observability matrix, but we can no longer bound $\kappa(\mathcal{O})$ via the eigengap.

To summarize, increasing the observation dimension (1) strengthens eigenvalue placement, (2) adds $\sqrt{d_{\mathcal{Y}}}$ factors to many of the norms, and (3) creates a difference between the max and min singular values of $C$, which will show up in the $\|L_\star\|_F \cdot \|C\|_F$ terms in the regret bounds. It is difficult to get control on the conditioning of $C$ in our Markov chain construction, but morally everything stated remains true.

## G. Theoretical Subtleties

In this section, we discuss a few nuances in our theory. Some appear paradoxical at first glance, but with more care they yield useful information and intuition about these proof techniques.

**Strong stability.** To begin, since we can construct the observer system with whichever poles we want, one could ask "why not just make $A - LC$ to be $\gamma$-stable for $\gamma = \frac{1}{2}$ and use $\widetilde{\Theta}\left(\frac{1}{\gamma}\right)$ regression terms?". The answer is that the number of autoregressive terms needed also scales with the spectral condition number of the resulting observer system, i.e. one would actually need $\widetilde{\Theta}\left(\frac{\log Q_\star}{\gamma}\right)$ many regression terms. There is a price to pay for more extreme eigenvalue placement given by $Q_\star$, and the cost of a fully stable observer system is seen by the number of regression terms needed to learn it. In the regimes where our loss bounds are nontrivial, pole placement is only moving a few eigenvalues by a large distance (and perhaps many by a small distance).

**Use of observations.** Next, since we use pole placement for two purposes (to change the system eigenstructure and to construct an LDS over observations against which to apply spectral filtering), one could ask "what value of $L$ should be

taken if the system already has desirable spectral structure?". For example, if we start with a symmetric autonomous LDS $x_{t+1} = Ax_t$ and $y_t = Cx_t$, there is no immediate input-output sequence-to-sequence mapping against which spectral filtering should hope to compete against. The classical way to enforce real-diagonalizability and marginal stability of $A - LC$ in this case would be to choose $L = 0$, but this does not fix the problem that there is no natural $[1, \alpha, \alpha^2, \ldots, \alpha^T]$ structure to an input-output mapping. The solution is to recognize that spectral filtering competes against *all* observer systems – for the choice of $L = 0$ we pay the full price of the initial state, and if $A$ is marginally stable we cannot hope for any decay. However, one can choose $L$ nonzero but such that $A - LC$ remains desirable (for example, if $A$ has an eigenvalue strictly less than 1 we can move it and only it, whereas if $A$ has all eigenvalues 1 then we can take $L$ proportional to $C^\top$). In this way, we construct an observer LDS with a nontrivial input-output mapping which retains the desirable spectral structure, and spectral filtering can efficiently learn to represent this.

**Spectral gap assumption.** In Theorem 4.1 we showed that if $Q_\star = \lim \sup_{\varepsilon \downarrow 0} Q_\star(f, h, \varepsilon, \Sigma)$ with $\Sigma = [0, 1 - \rho] \cup \{1\} \cup \mathbb{D}_{1-\gamma}$ then Algorithm 1 has loss bound

$$\sum_{t=1}^{T} \|\hat{y}_t - y_t\|^2 \leq O\left(R^2 \cdot \frac{\operatorname{poly} \log(Q_\star, T)}{\rho}\right).$$

To do so, we constructed a linear observer system with eigenvalues in $\Sigma$ and non-normality $Q_\star$ which reproduces the observation sequence.

Ignoring the $\mathbb{D}_{1-\gamma}$ strongly stable eigenvalues, consider an LDS $(D, C, x_0)$ with $D$ diagonal and $\sigma(D) \subset [0, 1 - \rho] \cup \{1\}$ which produces the observations

$$x_{t+1} = Dx_t, \quad y_t = Cx_t.$$

By simply predicting $\hat{y}_t = y_{t-1}$ at each step we incur loss

$$\sum_{t=1}^{T} \|y_{t-1} - y_t\|^2 \leq 2R \sum_{t=1}^{T} \|C(D^{t-1} - D^t)x_0\| \lesssim R^2 \sum_{t=1}^{T} \rho^t = O\left(R^2 \cdot \frac{1}{\rho}\right)$$

since the 1-eigenspace is killed by taking the difference.

Based on this fact, one could reasonably ask "if the signal $(y_t)_{t=1}^{T}$ is (approximately) generated by an observer system with spectrum in $\Sigma$, won't we get the same $O(R^2/\rho)$ loss as that of Theorem 4.1 by trivially predicting the previous observation?". To resolve this, we note that the theoretical comparator observer system is not diagonal: it is non-normal with diagonalizing condition number $Q_\star$. By reproducing the above argument, the trivial predictor which predicts the prior observation incurs loss

$$\sum_{t=1}^{T} \|y_{t-1} - y_t\|^2 \leq O\left(R^2 \cdot \frac{Q_\star}{\rho}\right).$$

The benefit of Theorem 4.1 (and indeed the effect of the learning performed by Algorithm 1) is to reduce this dependence to polylogarithmic in $Q_\star$.

# H. Other Experiments

In this section, we supplement the experimental results of Section 7. First, we demonstrate that spectral filtering over observations allows for the learning of asymmetric LDS with process noise. Afterwards, we experiment on some more nonlinear sequence prediction tasks.

## H.1. Linear Systems

Recall Algorithm 1, which performs spectral filtering over the observation history $y_t$ as well as the open-loop controls $u_t$. If we run the algorithm with $N^t \equiv 0$ and $m = 1$ then we essentially recover the original vanilla spectral filtering algorithm of (Hazan et al., 2017), which has a guarantee for symmetric LDS with no process noise. We will refer to this baseline as SF, and we will denote the full Algorithm 1 by SF+obs. For simplicity, we will only compare these two methods to highlight the contributions of this work; for comparisons between spectral filtering-based algorithms and other LDS learning methods, please see (Liu et al., 2024) or (Shah et al., 2025).

Our linear guarantee of Theorem A.3 proves two main benefits of the additional filtering over observations: (1) robustness to adversarial process noise and (2) the ability to learn asymmetric LDS. To demonstrate these advancements, we consider signals $(y_t)_{t=1}^T$ generated by an LDS according to (2.1) in two settings:

(1) $A$ is a Gaussian random $128 \times 128$ matrix, normalized for $\|A\| = 1$, and $w_t = 0.1 \cdot \sin(3\pi t) \cdot [1, \dots, 1] \in \mathbb{R}^{128}$ are correlated sinusoidal disturbances.

(2) $A$ is the $16 \times 16$ cyclical permutation matrix and $w_t = 0$.

In both cases, $d_{in} = d_{out} = 1$, we sample $B, C, x_0, (u_t)_{t=1}^T$ as i.i.d. Gaussians[16], we use the `optax.contrib.cocob` optimizer for simplicity[17], and we run Algorithm 1 with $m = 1$ and $h = 24$. The instantaneous losses are plotted in Figure 3, averaged over random seeds and smoothed.

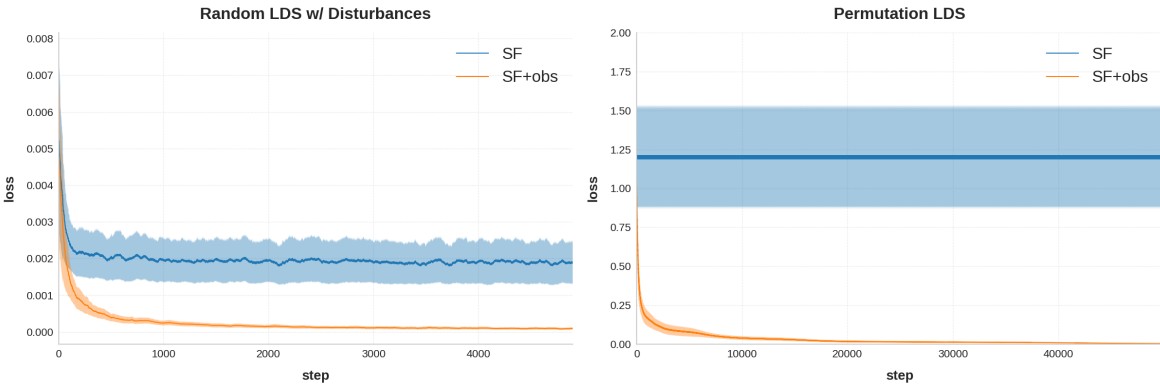

*Figure 3.* Instantaneous losses $\|\hat{y}_t - y_t\|^2$ plotted as a function of $t$ for experiments (1) and (2), averaged over 12 random seeds with a smoothing filter of length 100 and 1000, respectively. Shading indicates $\pm 1$ standard deviation.

As expected, filtering over observations improves the tolerance to asymmetry and correlated process noise, and on setup (1) the minimal loss achieved is a fixed deterministic nonzero value determined by the noise structure. On setup (2), we see that vanilla spectral filtering is unable to make any progress for the permutation system (which is the most asymmetric marginally stable LDS); by contrast, the simple addition of filtering over observations enables learning. Furthermore, in (2) the number of steps needed to achieve 0 loss is very large[18], as predicted by the value of $Q_\star$ (see Example D.5). Note that even though the systems in (1) are asymmetric, it is in a much tamer way that results in smaller $Q_\star$ and faster learning. To sum up, the benefits of filtering over observations appear to be exactly as predicted by Theorem A.3.

### H.2. Nonlinear Experimental Setup

Consider an autonomous nonlinear system, which generates a signal $(y_t)_{t=1}^T$ according to (1.1). The task is next-token prediction, in which we use $y_1, \dots, y_t$ to form predictions $\hat{y}_{t+1}$.

Imagine running Algorithm 1 with $J_t, M_t \equiv 0$, which corresponds to spectral filtering with regression over the observation history. As before, we run this with $h = 24$ and $m = 1$ (yielding $(h + m) \cdot d_y^2$ many parameters), and for optimizer we use either COCOB or ONS with tuned learning rate, depending on computational restrictions. For the following experimental results, we refer to this as SF. As baselines, we consider:

- Directly learning an observer LDS $(A, B, C, x_0)$ via online gradient descent on the next-token MSE loss, with

---

[16]For a marginally-stable system like the permutation LDS, if $u_t \sim \mathcal{N}(0, 1)$ then we are expecting $\|y_t\| \asymp \sqrt{t}$. To preserve boundedness of the observations, we decay the size of controls for the permutation experiment.

[17]Since the involved optimizations are convex, these results are robust to optimizer choice. By contrast, learning either of the above problems with another approach such as S4 or Mamba are nonconvex and very dependent on the optimizer used.

[18]Interestingly, for the permutation system there is an initial phase of rapid decrease, followed by a very slow second phase of refinement. Anecdotally, we find that for the first order optimizers we experimented with, the length of the initial phase appears independent of $d_h$ whereas the length of the second phase is governed by $Q_\star$. This indicates that perhaps other optimization algorithms would achieve faster convergence, which we leave to future work.

predictions of the form

$$\hat{y}_{t+1} = \sum_{j=0}^{t} CA^j B y_{t-j} + CA^{t+1} x_0.$$

We refer to this baseline as `LDS`, where we fix $d_h = 64$ to be the hidden dimension of the LDS. This predictor has $d_h^2 + 2d_h \cdot d_{\mathcal{Y}} + d_h$ parameters, and the involved optimization is nonconvex.[19]

- Performing an online variant of the eDMD algorithm (Williams et al., 2015), which fixes a dictionary $\{f_j\}_{j=1}^n$ of nonlinear observables $f_j : \mathcal{Y} \to \mathbb{R}$, defines the lifted states $z_t = [f_1(y_t), \ldots, f_n(y_t)] \in \mathbb{R}^n$, and attempts to fit a predictor $z_{t+1} = Az_t$, $\hat{y}_t = Cz_t$ for matrices $A \in \mathbb{R}^{n \times n}$, $C \in \mathbb{R}^{d_{\mathcal{Y}} \times n}$. We use the `pykoopman` library (Pan et al., 2023) implementation, which fits $A, C$ using least squares linear regression over the history of past observations. For the choice of observables, we let $n = d_{\mathcal{Y}} + 20$, where

$$f_j(y) = \begin{cases} y_j & j = 1, \ldots, d_{\mathcal{Y}} \\ \mathrm{RBF}_j(y) & j = d_{\mathcal{Y}} + 1, \ldots, n \end{cases},$$

where $\mathrm{RBF}_j$ refers to a radial basis function using thinplate splines with centers fitted to the data.[20] In other words, our nonlinear dictionary concatenates the initial observations with 20 RBF observables. We refer to this baseline as `eDMD`, and fit every $T/10$ steps for efficiency. This method has $n^2 + n \cdot d_{\mathcal{Y}}$ parameters.

- Lastly, we introduce a new method of eDMD combined with spectral filtering, which we call `SFeDMD`. We fix the same set of observables as in the `eDMD` baseline, but instead of fitting $A, C$ directly we apply the predictor from Algorithm 1 (with $h = 24$, $m = 1$) to learn the lifted dynamics. One could learn the parameters $N^t$ with either gradient descent or least squares linear regression: we choose the latter to have a more direct comparison against the `eDMD` baseline. This method has $(h + m) \cdot n \cdot d_{\mathcal{Y}}$ parameters.

To summarize, the algorithms we run consist of `SF` (which directly applies the observation-only version of Algorithm 1 using gradient descent), `LDS` (which attempts to directly learn an observer LDS via gradient descent), `eDMD` (which performs the classical eDMD algorithm via linear regression with a fixed dictionary of nonlinear observables), and `SFeDMD` (which uses the same dictionary of nonlinear observables and learns the dynamics improperly via spectral filtering and linear regression). These four algorithms constitute both proper and improper learning methods on both direct observations and nonlinear liftings. Rather than using online optimization algorithms for all methods, we choose to use regression for both eDMD baselines in order to directly apply the `pykoopman` library and for more direct comparison against eDMD experimental results of other works.

### H.3. Double Pendulum

A nonlinear system of physical interest is the double pendulum (a pendulum with two joints), which also exhibits a chaotic behavior and can have complicated marginally stable dynamics. We use the implementation in the Brax (Freeman et al., 2021) library, which is a `jax`-based physics simulator – this double pendulum is attached to a cart that can move horizontally on a rail, and it takes inputs which push the cart to the left or right. A fixed deterministic agent/policy operating in such an environment yields a deterministic nonlinear dynamical system which produces a sequence of states $x_0, \ldots, x_T$; for this environment, we define the state to be (the sines and cosines of) the angles and the angular velocities of both joints, together with the position and velocity of the cart to total $d_{\mathcal{X}} = 8$. We consider a fixed linear feedback controller of the form $u_t = Kx_t$ for a fixed $K \in \mathbb{R}^{1 \times 8}$ with i.i.d. standard Gaussian entries. As before, we can construct the sequence prediction task with either full observations $y_t = x_t$ or partial observations $y_t = Cx_t$ for a fixed $C \in \mathbb{R}^{1 \times 8}$ with i.i.d. standard Gaussian entries. Losses for both settings are plotted in Figure 4, averaged over random seeds and smoothed.

In some sense this is is a more difficult system than the Lorenz attractor, since a predictor needs to also learn the effects of $K$. Despite this, we see a similar story as in the Lorenz system: with full observations both `eDMD` and `SF` are able to predict well, and `SFeDMD` is able to predict very well. Furthermore, `SF` and `SFeDMD` are able to succeed when given

---

[19]In fact, the `LDS` baseline tends to blow up unless trained with the Adam optimizer with tuned learning rate, which we use for this baseline throughout the nonlinear experiments.

[20]This choice is described in Section 3 of (Williams et al., 2015), and is also used in the eDMD tutorials for the `pykoopman` library, available online. Furthermore, this dictionary seemed to be a good choice uniformly over all our tasks.

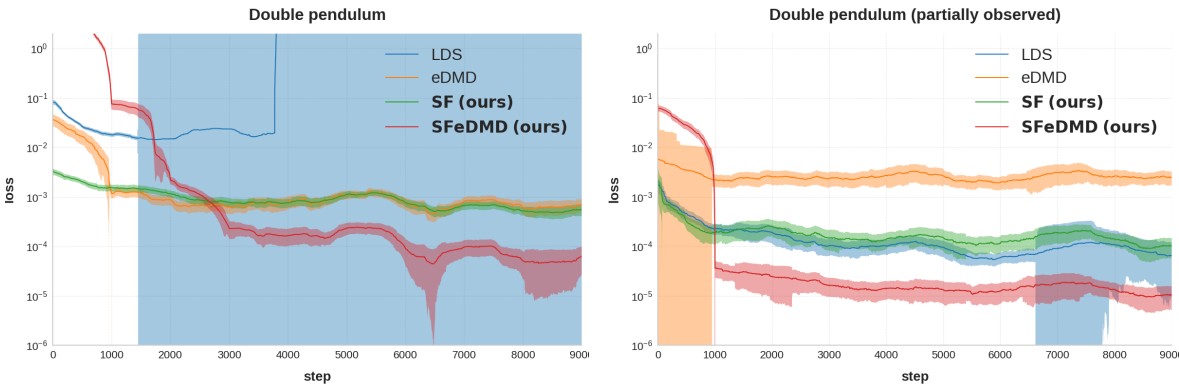

*Figure 4.* Instantaneous losses $\|\hat{y}_t - y_t\|^2$ plotted on log scale as a function of $t$ for the fully and partially observed double pendulum, respectively, averaged over 12 random seeds with a smoothing filter of length 1000. Shading indicates $\pm 1$ standard deviation.

partial observations, which `eDMD` struggles with. The `LDS` baseline is brittle – on some trajectories it achieves the same performance as `SF`, but on others it can completely diverge even at reasonable learning rates.

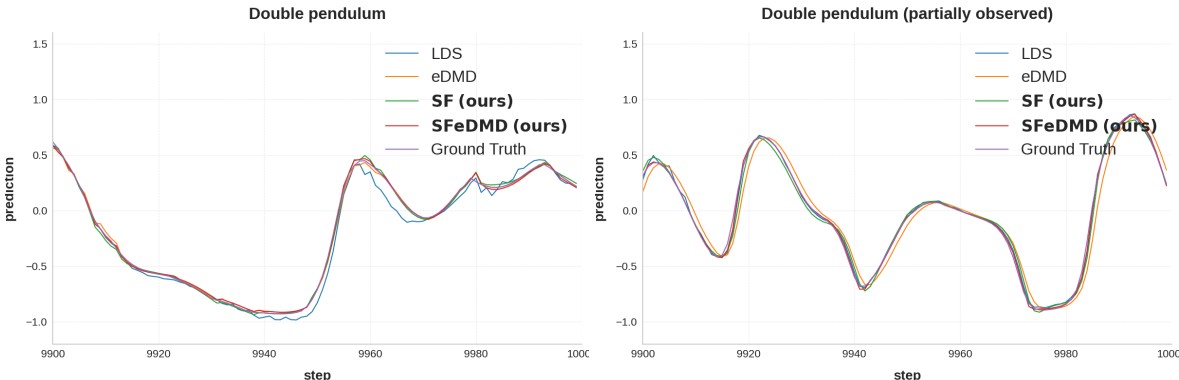

*Figure 5.* Predictions $\hat{y}_t$ of the four algorithms along with the ground truth $y_t$ plotted as a function of $t$ for the fully and partially observed double pendulum, respectively, on a single random seed. We zoom in on the last 100 steps of the trajectory. For the full observation case, since we cannot plot the full state we choose to plot the sine of the angle between the two arms of the pendulum, which is a representative coordinate for the task of next-state prediction.

To add a bit more clarity to the behaviors of these algorithms, we plot in Figure 5 the actual predictions $\hat{y}_t$ made during training at the end of a typical trajectory, along with the ground truth signal $y_t$. We see in the full observation setting that the pendulum is swinging irregularly in a way that the models can predict with reasonable accuracy (except for `LDS`, which suffers from unstable optimization). In the partial observation setting the algorithms all do quite well, but we do get to see some differences between the algorithms – `eDMD` is unable to be very precise due to the partial observation setting and appears to be predicting $\hat{y}_{t+1} = y_t$, whereas `SFeDMD` produces very precise predictions.

### H.4. Langevin Dynamics

As another autonomous nonlinear system, we consider stochastic gradient flow/Gibbs energy minimization/Langevin dynamics. For a clear and in-depth description of this dynamical system from a stochastic differential equation (SDE) perspective, see Section 1 of the textbook (Chewi, 2025). Given a differentiable loss/energy/potential function $V : \mathbb{R}^{d_x} \to \mathbb{R}$, one can define a dynamical system according to the SDE

$$dX_t = -\nabla V(X_t)dt + \sqrt{2}dW_t, \quad X_0 = x_0,$$

which produces a random sequence of states $X_1, \ldots, X_T$ (we use capital letters here to denote random variables). This is very common in the physical sciences and more recently in machine learning, and can be interpreted through many different perspectives:

- A first perspective is to interpret this as a **gradient descent**: we can imagine a discretized version via[21]

$$X_{t+1} = X_t - \eta \nabla V(X_t) + \sqrt{2\eta} w_t,$$

where $\eta$ is a small step size and $w_t \sim \mathcal{N}(0, 1)$ is an independent noise term that can be viewed as a stochastic gradient. In this way, the states correspond to the iterates of a first-order optimization algorithm.

- Under very mild assumptions, the distribution of $X_T$ for very large $T$ will approach the probability distribution $\pi$ on $\mathbb{R}^{d_\mathcal{X}}$ with (unnormalized) density $e^{-V}$. A particular trajectory can be viewed as a way to transform $x_0$ into a sample from $\pi \propto e^{-V}$, which yields a **sampling** perspective.

- Langevin diffusions arise naturally in thermodynamics and statistical physics, in which case $V$ is the energy of a particle/configuration – it is known that the equilibrium distribution (at least at temperature 1) will be the stationary distribution $\pi \propto e^{-V}$, which is sometimes also called the Gibbs distribution. The evolution of the physical system will follow these trajectories, yielding a **statistical physics** perspective.

In any case, it is clear that prediction of such systems is a useful task. The main difficulty is this: it can take very long for the system to equilibrate, and the transient/nonequilibrium dynamics are often the most useful (predicting optimization trajectories becomes useless if we wait until convergence). For some intuition, the Koopman operator[22] may have multiple marginally stable modes, in which case there is slow convergence (i.e. no strong mixing), and in such cases prediction is difficult. Under extra assumptions (strong convexity of $V$, a Poincare inequality for $\pi$, etc.) we would have exponentially-fast convergence mixing, but the general case is most interesting.

A crucial property of such SDEs is that they satisfy a detailed balance condition, and so the Koopman operator is a self-adjoint operator on $L^2(\pi)$ (see Remark 1 of (Kostic et al., 2022)). In our language, this implies that **there exists a well-conditioned high-dimensional symmetric LDS approximating the dynamics** – this is very good news for us, since it implies a small $Q_\star$ and so strong performance of Algorithm 1 due to Theorem 4.1.

To make the prediction problem difficult, we consider a potential on $\mathbb{R}^{d_\mathcal{X}}$ with exponentially-many asymmetric wells along each coordinate and a nontrivial coupling between coordinates, given by

$$V(x) = \sum_{j=1}^{d_\mathcal{X}} \left(0.05 \cdot x_j^4 - x_j^2 + 0.1 \cdot x_j\right) - 0.2 \cdot \sum_{1 \le i < j \le d_\mathcal{X}} x_i^2 x_j^2.$$

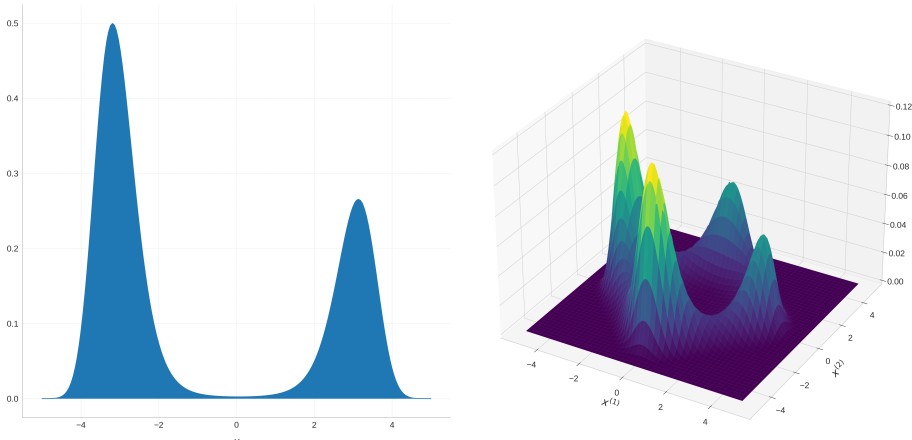

*Figure 6.* Densities of the stationary distribution $\pi$ corresponding to the chosen potential $V$ for $d_\mathcal{X}$ equal to 1 and 2, respectively. We see a number of asymmetric wells that is exponential in the dimension.

---

[21]This simple integrator with $\eta = 0.01$ is how we will numerically simulate SDEs for our experiments.

[22]Technically one would define the Koopman operator as a family $K_t$ sending $h(x) \to \mathbb{E}[h(X_t) \mid X_0 = x]$, and the derivative of $K_t$ is the generator of the SDE. The main story is the same – approximating the nonlinear dynamics by infinite-dimensional linear ones.

For convenience, we plot the density of the stationary distribution $\pi \propto e^{-V}$ in one and two dimensions in Figure 6. In Figure 7 we plot the training losses for the $d_{\mathcal{X}} = 64$ instantiation of the problem, averaged over random seeds and smoothed – we find that the `LDS` method consistently fails in the same way, suggesting some obstacle for learning an observer system via gradient descent on this task. By contrast, the other methods do well, and in this setting we find that the addition of the nonlinear dictionary offers no help. The minimal loss value reached is consistent with the size of the noise at each step (which is typically of size 0.02), agreeing with the lower bound of Proposition E.1. We anticipate more interesting behaviors for more complicated Langevin dynamics, experimentation on which we leave for future work.

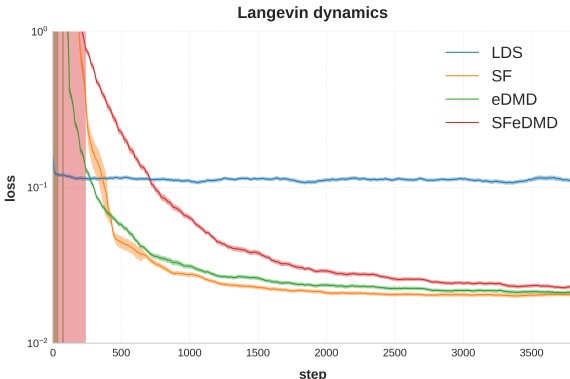

*Figure 7.* Instantaneous losses $\|\hat{y}_t - y_t\|^2$ plotted on log scale as a function of $t$ for the Langevin dynamics, averaged over 12 random seeds with a smoothing filter of length 200. Shading indicates $\pm 1$ standard deviation.

### H.5. Numerical Tests

Thus far, we have demonstrated the behaviors of Algorithm 1 on various nonlinear systems and compared to some simple baselines; we saw that spectral filtering over observations seems to succeed without much issue, and without the very slow optimization behaviors that were present in the permutation system (recall Figure 3). There is the intuition developed in the previous section that physical systems ought to have (approximately) real Koopman spectrum and therefore should be amenable to learning via spectral filtering. However, it would be nice to probe this question numerically, which we will quickly do in this section.

A classical approach to estimating Koopman eigenvalues of a nonlinear system is to run the eDMD algorithm as we have been doing, in which one learns a linear dynamics $A$ on a lifted space. By inspecting the eigenvalues of $A$, we gain understanding of the structure of (a finite-dimensional approximation to) the Koopman operator.[23] We take the `eDMD` models[24] trained on the full observation versions of our three nonlinear systems and plot the learned eigenvalues in Figure 8. As we can see, on the Langevin dynamics (which satisfies detailed balance) we have approximately real eigenvalues, whereas on the Lorenz and double pendulum systems (which exhibit chaotic behavior) we have more complex eigenvalues.

In addition, this numerical experiment allows us to compute an estimate for the spectral gaps: if we sort $1 = |\lambda_1| > |\lambda_2| \geq \ldots$, the spectral gap is given by $\rho = 1 - |\lambda_2|$. We give the empirical spectral gaps for these systems below:

$$\rho_{\text{Lorenz}} = 0.00168, \quad \rho_{\text{pendulum}} = 0.00477, \quad \rho_{\text{Langevin}} = 0.000694.$$

The mixing time of such a system scales as $\frac{1}{\rho}$, and our loss bound of Theorem 4.1 implies that after $\frac{1}{\rho}$ steps we get vanishing prediction error. For the above systems, the mixing time is around 1000 steps; however, we see from our experiments that Algorithm 1's learning often begins immediately. In other words, *spectral filtering appears able to predict transient dynamics* in certain settings. This indicates that the dependence of our analysis on a spectral gap is probably not sharp, and leaves room for theoretical improvement. See Theorem 1 of (Marsden et al., 2025) for a careful analysis of how spectral filtering successfully covers the $[1 - T^{-1/4}, 1]$ spectral region in the context of length generalization.

---

[23]Technically, this spectral relationship is only rigorous under the assumption that the dictionary of nonlinear observables spans an invariant subspace of the Koopman operator. This assumption is required for most eDMD-based methods to work at all, and inspecting the eigenvalues of the eDMD model is commonplace in practice regardless. See Section 5.1 of (Brunton et al., 2021).

[24]One could attempt to do this with the observer systems learned directly via gradient descent in the `LDS` method. These appear to be placed generically in the unit disk, though this result is inconsistent and brittle, varying significantly across seeds. By contrast, `eDMD` learns the same system eigenvalues across seeds, which are the ones we report.

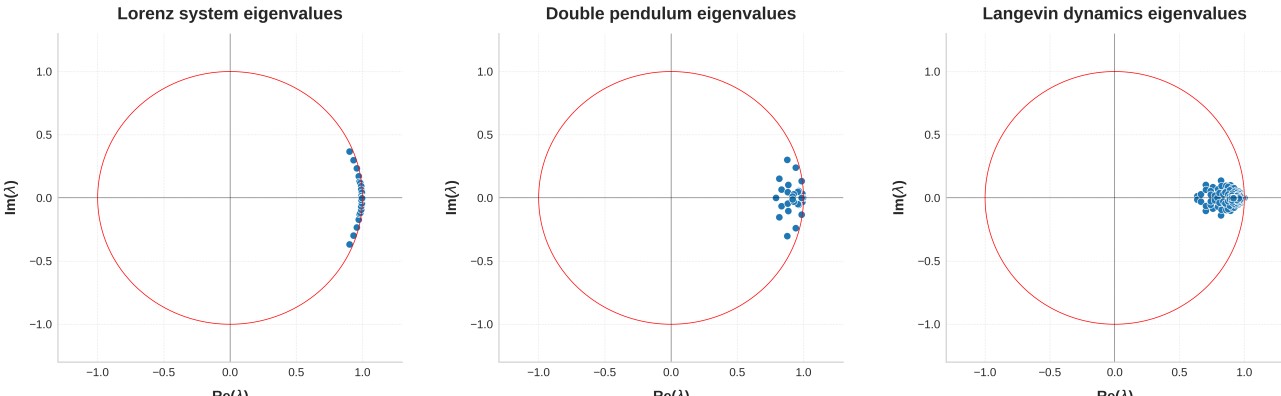

*Figure 8.* Eigenvalues of the lifted linear dynamics learned by the eDMD algorithm on the Lorenz system, double pendulum, and Langevin dynamics, respectively. The $x$ and $y$ axes display real and complex components, respectively, and we draw the unit circle in red for the reader's convenience.

