# OpenReview forum: "Universal Learning of Nonlinear Dynamics"
_ICML.cc/2026/Conference — ICML 2026 regular_

### Official Review · Reviewer_GKzU · 2026-02-17

**Soundness:** 3
**Presentation:** 3
**Significance:** 2
**Originality:** 2
**Overall Recommendation:** 3
**Confidence:** 4

**Summary:**

This paper presents Observation Spectral Filtering (OSF) to accurately predict a one-step-ahead prediction given the history of observations. Given the truncated history length $m$, a carefully chosen function using the eigendecomposition of Hankel matrix well-predicts the next observation. The authors present theorems that establish an upper bound on the sum of prediction errors (or squared prediction errors) in terms of $Q_\star$, a condition number of the approximate linear system.

**Compliance With Llm Reviewing Policy:**

Affirmed.

**Final Justification:**

Since no rebuttal was provided, I recommend rejection of this paper.

**Key Questions For Authors:**

1. It is understandable that the authors presented a bound on $\sum_{t=1}^{T} ||\hat y_t - y_t||^2$, but the final analysis, in my opinion, should be on the prediction error for the future observations; e.g.  $\sup_{t> T}||\hat y_t - y_t||$. Can the authors add a remark on this? One more question on this: since $O(\sqrt{T+1} ) - O(\sqrt{T})$ reduces, if $T \to \infty$, then does it mean that the prediction of the next observation is asymptotically exact, even if you fixed the autoregressive components $m$?

2. The authors’ analysis assume zero noise. When proving Theorem C.5, the authors assume an upper bound on the noise $W=0$ (see Line 1329) for Assumption A.1. This means that the authors’ method is prone to any types of disturbances; whenever $W\ne 0$, the additional error of $O(T)$ will be incurred for the prediction error bound, which is very restrictive. Can the authors propose the robustness of OSF to overcome at least for a specific type of noises? For example, noting that a linear dynamics can be extended to the case in which the functions $f$ and $h$ are Lipschitz (or simply focusing on the linear approximations),
 if the noise $w$ is additive and have zero-mean (or Martingale difference sequence) [1] and/or the nonzero noise $w\ne 0$ happens infrequently, e.g. probability less than $O(1/m)$ [2], then can you achieve $O(\sqrt{T})$ for the sum of prediction bounds?

My final recommendation will greatly rely on the response to Question 2.

[1] Oymak and Ozay, “Revisiting Ho–Kalman-Based System Identification: Robustness and Finite-Sample Analysis”, IEEE TAC, 2022.

[2] Kim and Lavaei, “System Identification from Partial Observations under Adversarial Attacks”, CDC 2025.

**Limitations:**

Yes.

**Strengths And Weaknesses:**

Strengths: The authors present rigorous theorems regarding various scenarios such as poly $\log(T)$ parameters, poly $\log(Q_\star T)$ parameters, or marginally stable complex systems. The error bound is on $O(\sqrt{T})$, which basically shows the decay of the prediction error as T increases. The experiments on Lorenz systems are very sound.

Weaknesses: The main weakness of this paper is on the absence of noise for both dynamics and observations. Although the analysis in the appendices consider the noises and the bounds include the noise terms, the final analysis does not include them.

---

### Official Review · Reviewer_QAuJ · 2026-03-04

**Soundness:** 4
**Presentation:** 4
**Significance:** 2
**Originality:** 3
**Overall Recommendation:** 4
**Confidence:** 5

**Summary:**

The paper studies online prediction for outputs generated by bounded, non-expansive non-linear systems. The authors show that such systems can be approximated by a high dimensional linear dynamical system (LDS) whose spectral properties can be controlled (in a desired way) through an observer based argument. Building on this reduction, they introduce Observation Spectral Filtering (OSF), which is an online learning algorithm predicting outputs (using spectral filters derived from Hankel matrices together with autoregressive feautures) and updates the predictor via online convex optimization (OCO). The main result establishes that OSF achieves vanishing average prediction error relative to the best LDS with suitable spectral structure , where the performance is governed by a conditioning parameter $Q_*$ measuring the difficulty of transforming the approximating LDS into a well-behaved one. Empirical experiments are provided to illustrate the predictive performance of the method on several dynamical systems benchmarks.

**Compliance With Llm Reviewing Policy:**

Affirmed.

**Final Justification:**

The reviews are divided, and the absence of an author response make the decision more difficult. However, I do not believe the lack of a rebuttal should by itself justify rejection, as the paper appears technically sound.

From my perspective, the main contribution is theoretical: the paper provides a nontrivial extension of spectral filtering guarantees from linear to bounded non-expansive nonlinear dynamical systems via an LDS approximation and an observer-based spectral conditioning argument. This is mathematically interesting and, in my view, correctly executed.

My concerns are primarily conceptual rather than technical. They are real but are not disqualifying defects, so they can ultimately be presented in camera-ready as open problems: the analysis  relies on nonconstructive objects (the high-dimensional LDS approximation and the observer transformation), and the guarantees hinge on the conditioning parameter $Q_*$, for which the paper does not provide a clear characterization or connection to observable properties of the system. In particular, it remains unclear when $Q_*$ is expected to be small, how typical such well-conditioned systems are, or whether this conditioning barrier is intrinsic (as suggested by example F.3). These issues make it difficult to build a structural understanding of the regime in which the framework is expected to succeed.

That said, I view these points as important questions rather than fatal flaws. The theoretical perspective introduced in the paper is interesting and could stimulate further work, especially if the role of $Q_*$ is better understood.

I would further note that I would be more strongly inclined toward acceptance if the authors explicitly acknowledged these unresolved issues and formulated them as a clear set of open problems.

**Key Questions For Authors:**

Question 1: While the method provides strong prediction guarantees, is there any interpretation of the learned parameters in terms of the underlying dynamical system, or is the approach inherently limited to black-box prediction?

Question 2: Many recent works in dynamical systems learning aim not only to predict trajectories but also to recover interpretable representations of the dynamics. Do the authors envision extensions of the OSF framework that could provide interpretable dynamical representations rather than purely predictive models?

Question 3: Can the authors quantify either structurally or in a probabilistic/topological sense, how typical the class of systems with well-conditioned observer realizations (small $Q_*$) is within the class of bounded non-expansive dynamical systems? This directly probes whether the theory applies to a large or small subset of systems.


Question 4: Are there structural properties of the underlying dynamics (like normality, contractivity, spectral separation or mixing conditions) that guarantee polynomial bounds on $Q_*$. A clearer characterization of such regimes would help understand when the theoretical guarantees are meaningful.


Question 5: In view of Example F.3 which already gives an explicit exponential lower bound for a concrete family. I wonder whether that phenomenon can be promoted to a systematic lethargy-style statement (with arbitrary slow\fast divergence), which would clarify how structural vs. worst case this conditioning barrier is. So, more specifically: Can the authors prove lower bounds showing $Q_*$ must be large over a class of non expansive observable systems, i.e. given any prescribed growth $b_n \to \infty$, construct a family for which $Q_*\geq b_n$ for the relevant $\Sigma$, proving that the framework is inherently non-universal?

**Limitations:**

The paper doe snot discuss any potential societal impacts. However, given that the work is primarily theoretical and focused on prediction for dynamical systems, no immediate negative societal impacts are apparent.

**Strengths And Weaknesses:**

Why significance: fair? The result is interesting theoretically, however its practical impact is unclear since the guarantee depends heavily on $Q_*$, which is not operatonally characterized. Additionally, the method relies on non-constructive objects (it does not give an interpretable representation of the nonlinear dynamics), so it does not improve interpretability. Furthermore, it extends the theory of spectral filtering (the main novelty of the paper lies in the theoretical extension to nonlinear dynamical systems via the LDS approximation and the observer-conditionig argument) but does not introduce a fundamentally new learning method (as the core pipeline used in the predictor, i.e. build Hankel spectral feautures and learn linear weights via online updates, in the paper is not completely new).

Strenghts: As I mentioned before the main contribution of the paper is theoretical. The authors provide a nontrivial theoretical extension of spectral filtering techniques from LDSs to bounded non expansive nonlinear systems, showing that prediction guarantees can still be obtained through an LDS approximation framework. The paper introduces a conditioning parameter $Q_*$ that quantifies the difficulty of transforming the approximating LDS into a form compatible with spectral filtering. Furthermore, the analysis provides online learning guarantees showing that the OSF algorithm achieves vanishing average prediction error relative to the best LDS with suitable spectral structure. Finally, their interdisciplinary perspective (dynamical systems, control theory, online learning and spectral methods) is interesting and may stimulate further work.

Weaknesses: From a mathematical perspective, the paper is difficult to interpret structurally. The learning algorithm ultimately reduces to a linear predictor over spectral features of the observation history, trained via OCO. The nonlinear system interpretation relies on an existential reduction to a high-dimensional LDS followed by spectral engineering via observer design, neither of which are constructed by the algorithm. As a result, the theoretical guarantees hinge on th conditioning parameter $Q_*$, but the paper does not provide a characterization of when $Q_*$ is small or how it relates to observable properties of the data generating system. This makes it challenging to build a conceptual understanding of the regime in which the framework is expected to succeed. Next, I am listing my main concerns (already covered in this small paragraph) and linking each of them to the specific questions I have to the authors.

Major Concern 1: As the LDS comparator is not explicitly constructed, the analysis relies on implicit objects whose relationship to the actual learned predictor is very difficult (or impossible) to interpret. This makes it challenging to understand the regimes in which the method is expected to succeed and limits the interpretability of the theoretical results. This directly motivates my Questions 1 and 2 below.

Major Concern 2: The guarantees depend on the existence of a well-conditioned observer realization measured by $Q_*$. From an operator-theoretic perspective, large classes of bounded and linear operators (like highly non-normal operators or hypercyclic dynamics in infinite dimensions that are topologically generic in many operator classes and in which cases $Q_*$ likely becomes enormous or undefined) do not admit such well-conditioned representations. Therefore, the theoretical scope of the result may exclude topologically large families of dynamical systems. This directly motivates my Question 3 below.


Major Concern 3: The theoretical guarantees rely on the existence of a well-conditioned (nonconstructive) observer realization measured by the constant $Q_*$. However, the paper provides limited insight into when $Q_*$ is small or how it can be estimated from data. In this regard, Lemma F.2 is not a checkable characterization we can estimate from data for a given nonlinear system. It looks more like an existence  style worst case bound in terms of $N$ and $vol (\Sigma)$. So this motivates my Question 4 below.


Major Concern 4: Lemma F.3 exhibits a natural and very structured LDS where the engineer spectrum into $\Sigma$ step can be catastrophically ill-conditioned, even though the system is not exotic at all. This directly motivates my Question 5 below.


Overall: While the paper raises several conceptual questions regarding the conditioning parameter $Q_*$, the nonconstructive nature of the LDS reduction and the interpretability of the resulting predictor, the theoretical contribution of extending spectral filtering guarantees to nonlinear dynamical systems is interesting and technical sound. Overall I lean toward Weak Accept!

---

### Official Review · Reviewer_2JG4 · 2026-03-07

**Soundness:** 3
**Presentation:** 3
**Significance:** 2
**Originality:** 3
**Overall Recommendation:** 3
**Confidence:** 3

**Summary:**

This paper proposes a spectral filtering–based algorithm for one-step prediction of unknown nonlinear dynamical systems and proves vanishing prediction error for bounded non-expansive systems using online convex optimization and a new control-theoretic notion of learnability.

**Compliance With Llm Reviewing Policy:**

Affirmed.

**Final Justification:**

Since my concerns have not been addressed, I will maintain my original score.

**Key Questions For Authors:**

1. What is the sampling rate for the experiment?
2. Why a nonlinear chaotic system can be modeled as a stable LDS?

**Limitations:**

yes

**Strengths And Weaknesses:**

Strengths:
1. The theoretical development appears technically solid and carefully derived.

2. The proposed OSF approach can learn stable linear dynamical systems with long-horizon memory, which is well suited for systems with near-marginal stability.

Weaknesses:
1. The phrase “universal learning” in the title is somewhat misleading. The paper focuses on one-step prediction, which represents only a limited subset of the broader time-series prediction problem.

2. The assumptions on the underlying nonlinear system appear restrictive. For example, it is unclear why the chaotic Lorenz system satisfies Assumption 3.1, since the butterfly effect implies that the composition $f\circ f \circ\cdots \circ f$ is generally not 1-Lipschitz.

3. The numerical evaluation is quite limited. The experiment is conducted on a very simple example, and the results would be more convincing if evaluated on more realistic datasets. In addition, measurement noise is not considered. It would be useful to understand how the proposed method performs on chaotic dynamics in the presence of noise.

4. The current experiment does not seem to provide a strong evidence why long-horizon memory matters. Does the short-horizon stable LDS have worse performance than the proposed approach?

5. In Line 120, the claim that noise can be removed without loss of generality is unclear. While noise can indeed be incorporated into the input via state augmentation, the resulting augmented system is generally not observable.

---

### Official Review · Reviewer_asQ4 · 2026-03-13

**Soundness:** 2
**Presentation:** 2
**Significance:** 2
**Originality:** 3
**Overall Recommendation:** 3
**Confidence:** 1

**Summary:**

The paper studies one-step prediction for an unknown nonlinear dynamical system from past observations. It proposes an improper learning approach that competes with a class of high-dimensional linear observer-like predictors, without explicitly identifying the hidden state or true nonlinear dynamics. It introduces Observation Spectral Filtering (OSF), proves vanishing prediction error for bounded, and provides control-theoretic analysis.

**Compliance With Llm Reviewing Policy:**

Affirmed.

**Final Justification:**

My main concerns are not addressed, and my score seem also in agreement with other reviewers.

**Key Questions For Authors:**

- How can one compute Q-star?
- In Lemma C.7 and definition 4.1, the discretization scale epsilon must scale with respect to 1/T^{3/2}. What is the consequence of this on scalability?
- Can you comment on the scalability of the proposed methods, in comparison to many existing system identification or prediction techniques?

**Limitations:**

Yes.

**Strengths And Weaknesses:**

The proposed Observation Spectral Filtering (OSF) as an improper learning method for predicting nonlinear dynamical systems is an interesting problem formulation.

In the paper’s derivation, it has an interesting way to apply control-theoretic tools for the prediction problem.

The paper is written in a way that some definitions come after they are used. It is not too easy to read without referring to the appendix, but this is understandable given the page limit.

The applicability of the techniques seems narrower than what is introduced in the introduction. In the introduction, the authors talk about the general scientific problems (econometrics, robotics, neuroscience, and language modeling), and about proposing a new paradigm for these scientific problems. However, the simulation is only performed for three-dimensional dynamical systems with three parameters. It is also unclear whether the problem used in the simulation is something that the large existing literature in the two existing paradigms (system identification and machine learning) has difficulty handling, or why this problem needs a new paradigm.  I feel this paper can be improved if the authors were to pick some examples of the applications listed in the intro, explain why such examples need a new paradigm, and why many existing tools in the two paradigms cannot be used for such examples.

---

### Decision · Program_Chairs · 2026-04-30

**Decision:**

Accept (regular)

**Comment:**

## Summary

The paper studies one-step prediction for an unknown nonlinear dynamical system from past observations. It proposes an improper learning approach that competes with a class of high-dimensional linear observer-like predictors, without explicitly identifying the hidden state or true nonlinear dynamics. It introduces Observation Spectral Filtering (OSF), proves vanishing prediction error for bounded, and provides control-theoretic analysis.

**The reviewers pointed out the following strengths:**

1. The proposed Observation Spectral Filtering (OSF) as an improper learning method for predicting nonlinear dynamical systems is an interesting problem formulation.

2. The paper provides a nontrivial extension of spectral filtering guarantees from linear to bounded non-expansive nonlinear dynamical systems via an LDS approximation and an observer-based spectral conditioning argument.

3. The paper introduces a new, theoretically grounded framework for prediction in nonlinear dynamical systems.

4. The paper addresses the challenging regime of partial observability, marginal stability, and adversarial noise in nonlinear dynamical systems.

5. **Overall:** The paper addresses a challenging problem, is novel and of significant interest to the machine learning community.

**The reviewers pointed out the following weaknesses:**

1. The experiments are performed with very simple three-dimensional dynamical systems with three parameters.

2. The learned predictor is not presented as an explicit state-space model of the underlying nonlinear dynamics.

## Final justification

All of the reviewers agreed that the paper is technically solid and well-motivated. The problem addressed by the paper is timely, and of significant interest to the machine learning community. Due to some misunderstanding, the authors were not able to upload the rebuttal on time. However, they sent their rebuttal to the AC, SAC, and PC. The AC is satisfied by the author’s response to the questions/concerns raised by the reviewers. The first two reviewers complain about the experiments being simplistic. However, I agree with the author that the goal of this work is not to outperform existing system identification or deep learning approaches empirically, but rather to introduce a new paradigm for prediction with guarantees under minimal assumptions. Importantly, the paper tackles the challenging regimes of partial observability, marginal stability, and adversarial noise, where classical approaches struggle.


The merits of the paper clearly outweigh its weaknesses.